# Autonomous and dynamic precursor selection for solid-state materials synthesis

Nathan J. Szymanski [1,2,5], Pragnay Nevatia[3,5], Christopher J. Bartel [4], Yan Zeng [1] ✉ & Gerbrand Ceder [1,2] ✉

Solid-state synthesis plays an important role in the development of new materials and technologies. While in situ characterization and ab-initio computations have advanced our understanding of materials synthesis, experiments targeting new compounds often still require many different precursors and conditions to be tested. Here we introduce an algorithm (ARROWS[3]) designed to automate the selection of optimal precursors for solid-state materials synthesis. This algorithm actively learns from experimental outcomes to determine which precursors lead to unfavorable reactions that form highly stable intermediates, preventing the target material's formation. Based on this information, ARROWS[3] proposes new experiments using precursors it predicts to avoid such intermediates, thereby retaining a larger thermodynamic driving force to form the target. We validate this approach on three experimental datasets, containing results from over 200 synthesis procedures. In comparison to black-box optimization, ARROWS[3] identifies effective precursor sets for each target while requiring substantially fewer experimental iterations. These findings highlight the importance of domain knowledge in optimization algorithms for materials synthesis, which are critical for the development of fully autonomous research platforms.

Conventional high-temperature synthesis based on solid-state reactions has long been used for the preparation of inorganic materials[1]. This method involves the mixing and subsequent heating of solid powders to facilitate reactions between them. Despite its apparent simplicity, the outcomes of solid-state synthesis experiments are often difficult to predict[2,3]. While density functional theory (DFT) calculations can be used to assess thermodynamic stability[4], even materials that are stable can sometimes be difficult to synthesize owing to the formation of inert byproducts that compete with the target and reduce its yield[5–8]. Further complicating matters is the prevalence of metastable materials[9] used in countless technologies including photovoltaics[10] and structural alloys[11]. Metastable materials are typically prepared using low-temperature synthesis routes, where kinetic control can be used to avoid the formation of equilibrium phases[12],

though recent work has shown that metastable phases can also appear as intermediates during high-temperature experiments[13–15]. To optimize the purity of a desired product, whether it be stable or only metastable, requires careful selection of precursors and reaction conditions. This selection process traditionally relies on domain expertise, reference to previously reported procedures for similar targets (if any exist)[16,17], and the use of heuristics such as Tamman's rule[18]. However, there is no clear roadmap to optimize the solid-state synthesis of novel inorganic materials, which can lead to many experimental iterations with no guarantee of success.

A new opportunity exists to accelerate inorganic materials development by leveraging computer-aided optimization to plan solid-state synthesis experiments, learn from their outcomes, and make improved decisions regarding the selection of precursors and

[1]Department of Materials Science and Engineering, UC Berkeley, Berkeley, CA 94720, USA. [2]Materials Sciences Division, Lawrence Berkeley National Laboratory, Berkeley, CA 94720, USA. [3]Department of Chemical Engineering, UC Berkeley, Berkeley, CA 94720, USA. [4]Department of Chemical Engineering and Materials Science, University of Minnesota, Minneapolis, MN 55455, USA. [5]These authors contributed equally: Nathan J. Szymanski, Pragnay Nevatia. ✉e-mail: yanzeng@lbl.gov; gceder@berkeley.edu

conditions that enable the formation of desired phases. Such an approach has found success in organic chemistry, where reactions can often be described by the breaking and formation of individual bonds[19,20]. This enables the use of retrosynthetic methods, which start from the target and work backward through stepwise reactions until a set of available starting materials is reached[21]. As many different reaction paths can lead to a given target, computer-aided optimization techniques based on Monte Carlo tree search and reinforcement learning have been successfully used to rapidly screen for promising synthesis routes[22–24]. In contrast, inorganic materials synthesis has yet to benefit from the widespread use of algorithms that can optimize experimental procedures. Their development is hindered by the difficulty of modeling solid-state reactions, where the corresponding phase transformations involve concerted displacements and interactions among many species over extended distances[2]. Some progress has been made in simplifying the analysis of solid-state reaction pathways by decomposing them into step-by-step transformations that take place between two phases at a time, hereafter referred to as pairwise reactions[6,15]. However, it remains difficult to predict the temperature at which a given pairwise reaction will occur, as well as what phase(s) will form as a result of that reaction.

To determine which reaction outcomes are most plausible for a given set of precursors and conditions, much of the existing work on computer-aided planning for solid-state synthesis has relied on the analysis of thermochemical data based on density functional theory (DFT) calculations[25,26]. For example, McDermott et al. introduced a graph-based approach that ranks various reaction pathways by a cost function designed to account for changes in the Gibbs free energy of reaction along each path[27]. A related approach developed by Aykol et al. parameterizes reactions by two axes—one that approximates the nucleation barrier of the targeted phase and another that accounts for its competition with possible byproducts—from which optimal reactants can be identified along the Pareto front[28]. Alternatively, machine learning models can be trained on synthesis data from the literature and applied to suggest effective precursors and conditions for a given target by considering its similarity with previously reported materials[16,17]. While these methods have been successfully applied in some cases, their use remains limited as they only provide a fixed ranking of synthesis routes for a given material, which is not readily updated should the initial experiments fail.

In the place of fixed ranking schemes, active learning algorithms have also been used for the optimization of synthesis procedures[29,30]. These algorithms can adapt from failed experiments and decide which parameters should be tested in later iterations. Bayesian optimization and genetic algorithms have found success when coupled with synthesis techniques based on flow chemistry[31] and thin film deposition[32]. However, these black-box approaches are often restricted to handling continuous variables such as temperature and time, while categorical variables are more difficult to optimize. For example, choosing which precursors to use for the synthesis of a novel material is particularly challenging as it involves discrete selections from a vast range of chemical compositions and structures, instead of simply fine-tuning parameters on a continuous scale. Recent work has made progress on this front by combining parallel synthesis experiments with tensor decomposition analysis, which can be used to predict the most effective starting materials and processing conditions from just a subset of their possible combinations[33].

In this work, we build upon existing methods to optimize solid-state synthesis procedures by incorporating physical domain knowledge based on thermodynamics and pairwise reaction analysis. This is accomplished using Autonomous Reaction Route Optimization with Solid-State Synthesis (ARROWS³), an algorithm designed to guide the selection of precursors for the targeted synthesis of inorganic materials. Given a desired structure and composition, ARROWS³ uses existing thermochemical data in the Materials Project to form an initial

ranking of precursor sets based on their DFT-calculated reaction energies[34,35]. Highly ranked precursors are suggested for experimental validation throughout a range of temperatures, which are iteratively probed and analyzed using machine learning algorithms to identify the intermediates that form along each precursor set's reaction pathway. When such experiments fail to produce the desired phase, ARROWS³ learns from their outcomes and updates its ranking to avoid pairwise reactions that consume much of the available free energy and therefore inhibit the formation of the targeted phase. To benchmark the performance of ARROWS³, we conducted 188 synthesis experiments targeting $YBa_2Cu_3O_{6.5}$, forming a comprehensive reaction dataset that critically includes both positive and negative results. Our approach is shown to identify all the effective synthesis routes from this dataset while requiring fewer experimental iterations than Bayesian optimization or genetic algorithms. We further demonstrate that ARROWS³ can be applied in line with experiments to guide the selection of precursors for two metastable targets, $Na_2Te_3Mo_3O_{16}$ and $LiTiOPO_4$, each of which were successfully prepared with high purity.

## Results
### Design of ARROWS³
The logical flow of ARROWS³ is summarized in Fig. 1 and detailed in the Methods section. Given a target material specified by the user, in addition to the precursors and temperatures that may be used for its synthesis, ARROWS³ forms a list of precursor sets that can be stoichiometrically balanced to yield the target's composition. In the absence of previous experimental data, these precursor sets are initially ranked by their calculated thermodynamic driving force ($\Delta G$) to form the target (Fig. 1a). While many factors influence the rates at which solid-state reactions proceed[36], those with the largest (most negative) $\Delta G$ tend to occur most rapidly[15,16,37]. However, such reactions may also be slowed by the formation of intermediates that consume much of the initial driving force[7]. To address this, ARROWS³ proposes that each precursor set be tested at several temperatures, thereby providing snapshots of the corresponding reaction pathway (Fig. 1b). The intermediates formed at each step in the reaction pathway are identified using X-ray diffraction (XRD) with machine-learned analysis[38]. ARROWS³ then determines which pairwise reactions led to the formation of each observed intermediate phase (Fig. 1c), and it leverages this information to predict the intermediates that will form in precursor sets that have not yet been tested (Fig. 1d). In subsequent experiments, ARROWS³ prioritizes sets of precursors that are expected to maintain a large driving force at the target-forming step ($\Delta G'$), i.e., even after intermediates have formed (Fig. 1e). This process is repeated until the target is successfully obtained with sufficiently high yield, as specified by the user, or until all the available precursor sets have been exhausted.

To validate the effectiveness of ARROWS³, new experimental synthesis data is needed. Existing results from the literature tend to be heavily biased toward positive results, which precludes the development of models that can learn from failed experiments[39]. We therefore built a solid-state synthesis dataset for $YBa_2Cu_3O_{6.5}$ (YBCO) by testing 47 different combinations of commonly available precursors in the Y–Ba–Cu–O chemical space, which were mixed and heated at four synthesis temperatures ranging from 600 to 900 °C. Importantly, this dataset includes both positive and negative outcomes, i.e., reactions that do and do not yield sufficiently pure YBCO. As such, it can be used as a benchmark on which to test ARROWS³ and compare its efficacy with alternative optimization algorithms. Two additional chemical spaces are also considered, where we use ARROWS³ to actively guide the experiments. The first set of experiments targeted $Na_2Te_3Mo_3O_{16}$ (NTMO), which is metastable with respect to decomposition into $Na_2Mo_2O_7$, $MoTe_2O_7$, and $TeO_2$ according to DFT calculations[40]. The second set of experiments targeted a triclinic polymorph of $LiTiOPO_4$ (t-LTOPO), which has a tendency to undergo a phase transition into a

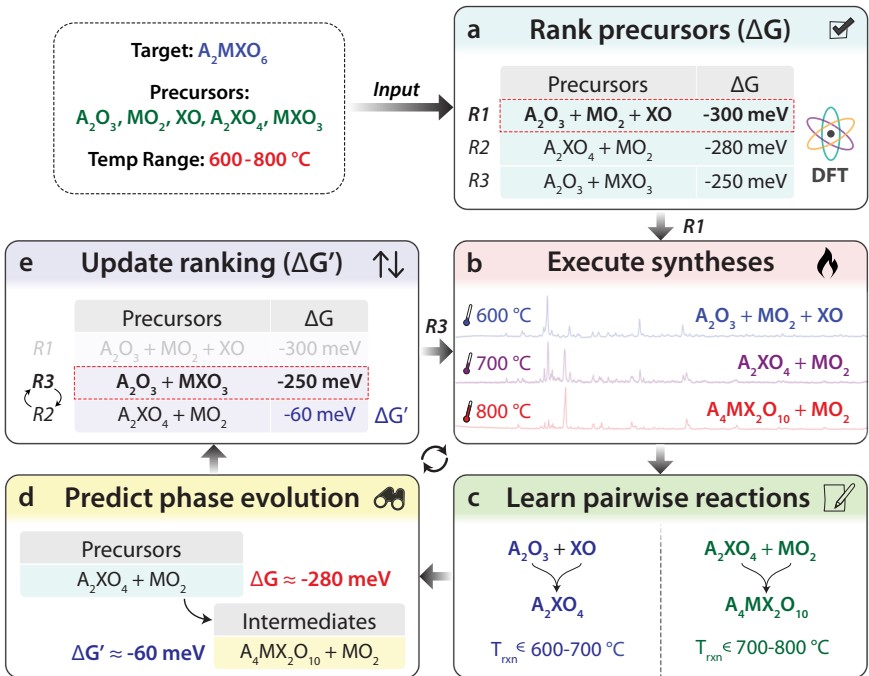

**Fig. 1 | A schematic illustrating how ARROWS³ guides precursor selection.**
**a** Reactions based on different precursor sets (R1, R2, and R3) are initially ranked by their driving force ($\triangle$G) to form the target, which is obtained from density functional theory (DFT) calculations. **b** Experiments are performed at iteratively higher temperatures to identify reaction intermediates. The chemical formulae listed in this panel represent the phases identified from X-ray diffraction (XRD) measurements. **c** Pairwise reaction temperatures ($T_{rxn}$) and products are gleaned from the experimental data. **d** Using the identified pairwise reactions, intermediates are predicted for other precursor sets and their remaining driving forces ($\triangle$G') are updated accordingly. **e** The precursor ranking is updated based on the newly calculated $\triangle$G'. All chemical formulae shown are placeholders for arbitrary compounds, and in general there is no restriction on the compositions where ARROWS³ is applicable.

lower-energy orthorhombic structure (o-LTOPO) with the same composition[41]. The features of each space tested are summarized in Table 1. Further details regarding the corresponding experiments are provided in "Methods".

## YBCO

Before discussing the optimization of YBCO synthesis using ARROWS³, we first summarize the outcomes from all 188 experiments to give context regarding the difficulty of obtaining high-purity YBCO while using a hold time of 4 h. Such a short hold time was used specifically to make the optimization task more challenging, as longer heating durations with intermittent regrinding are typically required to form highly pure YBCO samples[42]. Indeed, only 10 of the 188 experiments performed in the current work led to the formation of pure YBCO without any prominent impurity phases that could be detected by XRD-AutoAnalyzer[38]. Another 83 experiments gave partial yield of YBCO, in addition to several unwanted byproducts. Figure 2a shows the distribution of YBCO yield (wt. %) at each synthesis temperature sampled in this work. Generally, the use of higher temperature leads to increased yield of YBCO, likely due to enhanced reaction kinetics. Precursor selection also has a marked effect on the target's yield. Figure 2b shows the success rate of each precursor, which we define as

the percentage of sets where that compound was included and resulted in the formation of YBCO without any detectable impurities. This plot suggests that the less commonly used binary precursors tend to outperform their standard counterparts. For example, BaO and $BaO_2$ have moderately high success rates of 46% and 22%, respectively, whereas sets with $BaCO_3$ always produce impure samples (0% success rate). Precursor sets including $Y_2Cu_2O_5$ and $Ba_2Cu_3O_6$ also have comparably high success rates of 33% and 31%, respectively. We will later show that these ternary phases enable the direct formation of YBCO while circumventing inert byproducts such as $Y_2BaCuO_5$.

Figure 2c displays a pie chart containing the four most common impurity phases that coexist with YBCO, or prevent its formation entirely, at 900 °C. Each slice in the pie chart represents the fraction of experiments where the specified impurity phase appears. Most of the impure samples (28/44) contain $BaCuO_2$ or $Y_2BaCuO_5$, which are known to be relatively inert during the synthesis of YBCO, requiring intermittent grinding to improve the sample's purity[42,43]. CuO is another frequent impurity, though it only ever appears with at least one other byproduct that is Cu deficient. When such phases do not form, CuO contributes to the formation of YBCO, as evidenced by its success rate of 20%. The fourth most common impurity is $BaCO_3$, which is likely slow to react owing to its high decomposition temperature in air (1000 °C)[44,45]. We note that such information could in principle be leveraged when designing the search space, e.g., by removing $BaCO_3$ from the list of precursors since the proposed temperature range lies below its known decomposition temperature. Indeed, doing so reduces the number of experiments required to identify all optimal synthesis routes from 87 to 70 (Supplementary Fig. 1).

To determine whether ARROWS³ can effectively distinguish between successful and failed synthesis routes, we assessed how many iterations are required to identify all 10 optimal experiments that

**Table 1 | Information regarding three search spaces on which ARROWS³ was tested**

| Target | $N_{sets}$ | Temperatures (°C) | $N_{exp}$ |
|---|---|---|---|
| $YBa_2Cu_3O_{6+x}$ | 47 | 600, 700, 800, 900 | 188 |
| $Na_2Te_3Mo_3O_{16}$ | 23 | 300, 400 | 46 |
| t-$LiTiOPO_4$ | 30 | 400, 500, 600, 700 | 120 |

$N_{sets}$ and $N_{exps}$ represent the number of precursor sets and experiments, respectively.

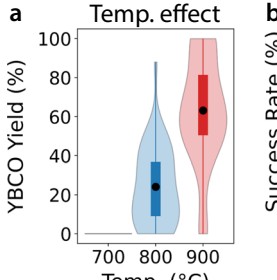

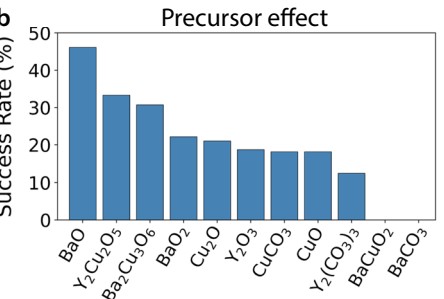

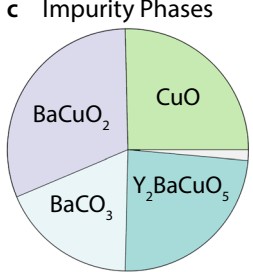

**Fig. 2 | A summary of outcomes from the synthesis experiments targeting YBa$_2$Cu$_3$O$_{6.5}$ (YBCO). a** Distributions of YBCO yield (wt. %) at different synthesis temperatures represented using violin and box plots, where each box extends from the lower to upper quartiles. **b** The success rate of each precursor, defined as the percentage of sets where that compound is included and forms YBCO without any impurities. **c** Common impurity phases that prevent YBCO formation are shown by a pie chart, where each slice represents the relative number of occurrences for each compound at 900 °C. The small gray sliver includes two less commonly observed impurities, YBaCu$_3$O$_7$ and YBa$_4$Cu$_3$O$_9$.

result in the formation of YBCO without any detectable impurities. While in practice it would be sufficient to identify just one optimal synthesis procedure for a given target, tasking the algorithm with identifying all optimal procedures for YBCO allows us to showcase its ability to learn over many experimental iterations (Supplementary Fig. 2). It also reduces the likelihood that ARROWS[3] discovers an optimal synthesis route by chance, thereby increasing our confidence in the performance of the algorithm.

As a baseline with which to compare the performance of ARROWS[3] on the YBCO dataset, we applied D-optimal design with progressively larger sets of proposed experiments. This approach aims to select the experiments whose outcomes will be maximally informative[46] to a model that maps the input variables (precursors and temperature) onto the output (YBCO yield). Here we assume a linear relationship between the two (Supplementary Note 1). We also applied two active learning algorithms, Bayesian optimization (BO) and a genetic algorithm (GA), to the same task by using a one-hot representation of each precursor (Supplementary Note 2). These algorithms are known to perform well on numerical inputs such as temperature[47,48]; however, their effectiveness with respect to categorical inputs is less well proven. To specifically probe the latter case, we constrained BO and GA to optimize the selection of precursors while sampling all temperatures for each precursor set. Both black-box algorithms have stochastic elements and were therefore applied to the YBCO dataset 100 times, each with a random starting seed, and their results were averaged. Because ARROWS[3] and D-optimal design are both deterministic algorithms, only a single run was performed to validate each on the YBCO dataset.

Figure 3a shows the number of optimal synthesis routes (those yielding pure YBCO) discovered with respect to the number of experiments queried by each algorithm. ARROWS[3] successfully identified all 10 optimal routes from 87 experiments, which account for just 46% of the entire design space (spanning 188 experiments). D-optimal design, on the other hand, required 165 experiments to accomplish the same task. However, it is worth noting that D-optimal design was quick to identify three optimal synthesis routes in the first 12 experiments. ARROWS[3], although slower to identify optimal routes in the early stages of optimization, eventually surpassed D-optimal design once it gathered sufficient information regarding the reactivity of various phases in the Y−Ba−Cu−O chemical space.

Both active learning algorithms performed poorly on the YBCO dataset, with BO and GA requiring on average 164 and 167 experiments to identify all ten optimal synthesis routes. We suspect the ineffectiveness of these algorithms is related to their use of one-hot representations for the precursors, which treat each compound independently and contain no physical information regarding their composition or structure. In contrast, ARROWS[3] encodes

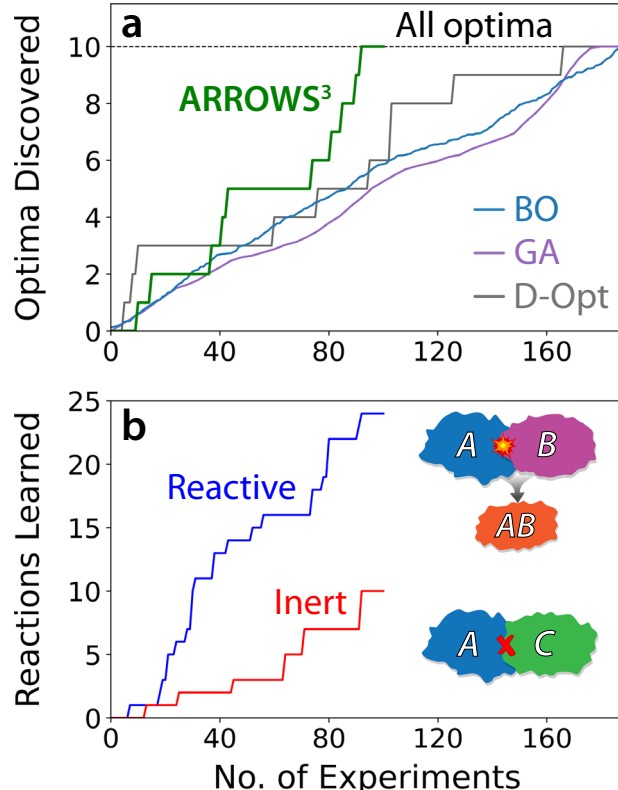

**Fig. 3 | Optimization results on the YBa$_2$Cu$_3$O$_{6.5}$ (YBCO) synthesis dataset. a** Number of optimal synthesis routes identified as a function of the experimental iterations required by ARROWS[3], Bayesian Optimization (BO), a Genetic Algorithm (GA), and D-Optimal design (D-Opt). The dashed line represents the total number of optimal synthesis routes in the dataset. **b** Pairwise reactions discovered by ARROWS[3] with respect to the number of experiments queried.

compositional and thermodynamic information in its optimization through its ranking by ΔG. It also learns from failed experiments to avoid pairwise reactions that form inert byproducts such as BaCuO$_2$ and Y$_2$BaCuO$_5$, instead prioritizing sets of precursors expected to retain a strong driving force (ΔG') to form YBCO.

Figure 3b displays the number of pairwise reactions learned by ARROWS[3] with respect to the number of experiments that were queried. This plot includes pairs of phases that react within the temperature range considered (≤ 900 °C), denoted reactive pairs, as well as the phases that do not react within that range, denoted inert pairs.

From 87 experiments, ARROWS[3] gained information regarding 34 pairwise interactions, including 24 reactive and 10 inert pairs. We find that the identification of new successful synthesis routes is often preceded by the discovery of new pairwise reactions. For example, ARROWS[3] learned from experiments 30–34 that BaO reacts with CuO to form $BaCuO_2$ at 800 °C, which subsequently reacts with $Y_2O_3$ at 900 °C to form $Y_2BaCuO_5$. Because these pairwise reactions consume much of the driving force that remains to form YBCO, the algorithm decides to prioritize sets of precursors that do not contain such reactive pairs (BaO|CuO or $BaCuO_2$|$Y_2O_3$). This decision leads to the successful discovery of three new synthesis routes that produce YBCO without any detectable impurities, as shown by the steep rise of the green curve in Fig. 3a between experiments 38-43. While previous work has shown that $BaCuO_2$ can effectively contribute to YBCO formation when it melts in combination with CuO[6], there was no evidence of melting in our samples owing to the use of low synthesis temperatures (≤ 900 °C) that ensured all products could be easily extracted.

In addition to learning which pairwise reactions should be avoided, ARROWS[3] also learns which reactions are beneficial to achieve high target yield. During the optimization of YBCO synthesis, it learned from experiments 72-80 that $BaO_2$ reacts with CuO to 700 °C to form $Ba_2Cu_3O_6$, which upon further heating to 900 °C reacts with $Y_2O_3$ to form YBCO. Accordingly, subsequent experimental iterations are chosen based on precursor sets that either include $Ba_2Cu_3O_6$ or are expected to form it as an intermediate phase. As shown in Fig. 3a, this leads to the rapid identification of all remaining experiments that successfully form YBCO shortly after the 80th experimental iteration.

To showcase the pairwise reactions learned by ARROWS[3], we present in Fig. 4 a heatmap where each square represents a pair of phases. If any information was learned regarding the reactivity of that pair, the square is colored a light shade of blue according to the temperature at which a reaction proceeds. If a pair was instead found to be inert at all temperatures ≤ 900 °C, a dark shade of blue is used. We also denote reactions that produce YBCO (yellow star) or its

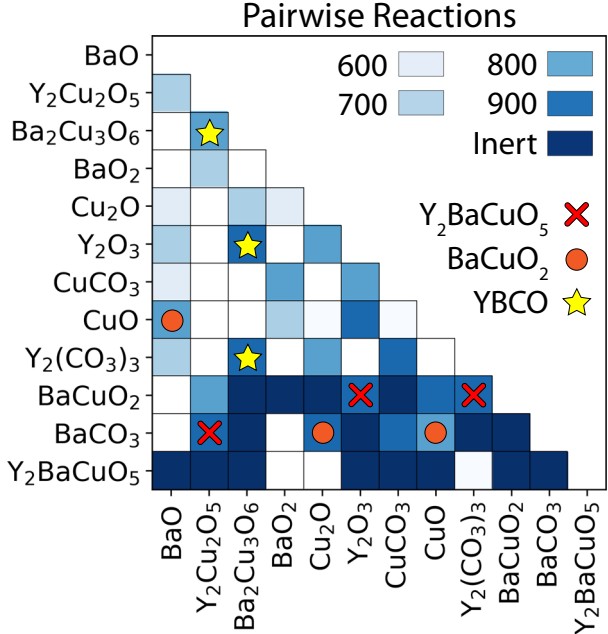

**Fig. 4 | Pairwise reactions in the Y–Ba–Cu–O chemical space, illustrated by a heatmap where the color of each square represents the temperature (°C) at which a reaction is observed.** Inert pairs correspond to phases that do not react within the temperature range considered. White squares (unshaded) represent pairs of phases whose reactivity was not learned by ARROWS[3]. Yellow stars denote pairs that react to produce $YBa_2Cu_3O_{6.5}$ (YBCO). Orange circles and red crosses denote pairs that form impurities, $Y_2BaCuO_5$ and $BaCuO_2$, respectively.

competing phases, $BaCuO_2$ (orange circle) and $Y_2BaCuO_5$ (red cross). This heatmap reveals that YBCO forms at 900 °C when $Ba_2Cu_3O_6$ reacts with $Y_2O_3$ or $Y_2(CO_3)_3$. It is separately observed that $Ba_2Cu_3O_6$ reacts with $Y_2Cu_2O_5$ when both are present at 800 °C, resulting in a mixture of YBCO and CuO. The direct formation of YBCO from $Ba_2Cu_3O_6$ and $Y_2Cu_2O_5$ provides an explanation as to why both phases have high success rates when used as precursors (Fig. 2b). In contrast, the 0% success rates associated with $BaCO_3$ and $BaCuO_2$ can be traced to the limited reactivity of each phase with many of the precursors tested here. This is illustrated in Fig. 4 by the dark-blue shading that signifies inert reaction pairs in the rows corresponding to $BaCO_3$ and $BaCuO_2$. Even when $BaCO_3$ does react, it often produces $BaCuO_2$ or $Y_2BaCuO_5$, which are both common impurity phases that preclude the formation of YBCO. The presence of $Y_2BaCuO_5$ is particularly detrimental to the synthesis of YBCO as it does not react with any precursor in the allotted hold time of 4 h, which ARROWS[3] learns over the course of the 87 experiments we performed. For a more detailed visualization of the information gleaned from each stage in the experimental process, we plot in Supplementary Fig. 3 an evolution of the heatmap displaying which pairwise reactions were learned after 30, 60, and 90 experiments.

There also exist some pairs of compounds whose reactivity was not learned by ARROWS[3] during its optimization of YBCO synthesis. These 23 pairs are denoted by the white (unshaded) squares in Fig. 4. We observe two factors that prevent ARROWS[3] from learning pairwise reaction information. First, when two phases (e.g., A|B) react in a three-phase set (A, B, and C), the algorithm is unable to learn how the remaining phase (C) interacts with the already reacted compounds (A and B). Separate experiments based on the individual pairs (A|C and B|C) would be required to determine their reactivity. Second, when multiple pairwise reactions take place within the specified temperature interval ($\Delta T = 100$ °C), the algorithm cannot determine the precise reaction sequence between the lower and upper temperatures (e.g., between 600 and 700 °C). In principle, the second limitation can be overcome by using a smaller temperature interval; however, doing so would also require more experiments.

The results presented in Figs. 3 and 4 were obtained by querying experiments in a serial (one-by-one) fashion. This allowed ARROWS[3] to continually learn from each experimental outcome and update its ranking of precursor sets accordingly. However, traditional experiments are often parallelized. For example, multiple sets of precursors with a shared synthesis temperature may be tested simultaneously by loading them into one furnace[33]. Such an approach is also compatible with ARROWS[3], for which a batch size can be specified to control how many experiments are suggested at each iteration. As shown in Supplementary Fig. 4, the use of a larger batch size reduces the number of iterations (i.e., batches) required to identify all the optimal synthesis routes for YBCO. This also leads to shorter hold times required in the furnace. However, because a larger batch size limits the opportunities where ARROWS[3] can learn and update its ranking, it also leads to a larger number of individual samples that must be queried to identify the optimal routes. The efficiency with which samples are queried becomes particularly affected at later stages in the experiments, where the algorithm has sufficient knowledge of the chemical space to make frequent updates to its ranking of different precursor sets. Hence, there exists a tradeoff between the number of batches and individual samples required to complete the optimization process, and the batch size acts as a hyperparameter to adjust this tradeoff depending on the user's objectives and experimental setup.

## NTMO

ARROWS[3] was next tasked with optimizing the yield of $Na_2Te_3Mo_3O_{16}$ (NTMO) by choosing from 23 different precursor sets and two synthesis temperatures (300 and 400 °C), which were kept low to avoid melting of the samples[49]. In the top panel of Fig. 5a, we show the weight

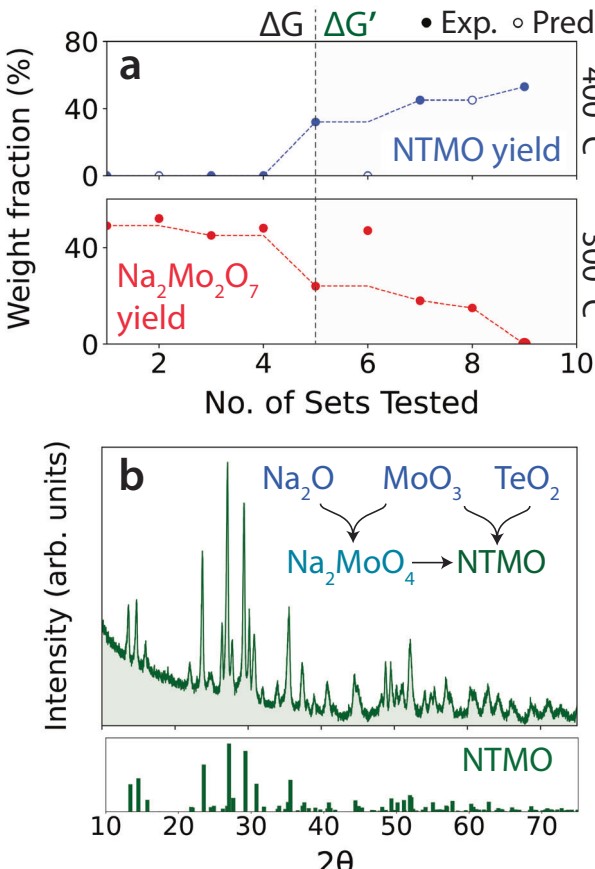

**Fig. 5 | Optimization of Na$_2$Te$_3$Mo$_3$O$_{16}$ (NTMO) synthesis using ARROWS$^3$. a** The top panel shows the weight fraction of NTMO obtained from each precursor set when tested at 400 °C. The bottom panel displays the weight fraction of a competing phase, Na$_2$Mo$_2$O$_7$, obtained at 300 °C. Solid (hollow) dots represent experimental (predicted) values. **b** X-ray diffraction (XRD) pattern measured from the product of the optimized precursor set, Na$_2$O + TeO$_2$ + MoO$_3$ after an 8 h hold at 400 °C. For comparison, a reference pattern is shown for NTMO (ICSD #171758).

fraction of NTMO obtained at 400 °C for each precursor set that was tested. The solid dots represent experimentally observed weight fractions, whereas the hollow dots represent predictions made based on the intermediates formed at 300 °C. As detailed in the Methods section, a precursor set occasionally produces identical intermediates phases to a previously explored set. In this case, higher temperatures do not require sampling since their outcomes can already be predicted based on previous synthesis outcomes.

None of the four initial precursor sets, which were selected based on their DFT-calculated reaction energies (ΔG), produced any detectable amount of the target. Their failures can be attributed to the formation of an intermediate phase, Na$_2$Mo$_2$O$_7$, that consumes much of the available free energy and precludes the formation of NTMO. This is confirmed by the thermodynamically unfavorable driving force (i.e., positive reaction energy) associated with NTMO formation based on the hypothetical reaction between Na$_2$Mo$_2$O$_7$ and two commonly used precursors, TeO$_2$ and MoO$_3$:

$$Na_2Mo_2O_7 + 3TeO_2 + MoO_3 \rightarrow Na_2Te_3Mo_3O_{16}(\Delta G' = +13\,meV/atom)$$

To further illustrate the limiting effect that Na$_2$Mo$_2$O$_7$ has on the formation of NTMO, we plot in the bottom panel of Fig. 5a the weight fraction of Na$_2$Mo$_2$O$_7$ obtained at 300 °C for each precursor set, revealing a clear tradeoff between the yield of this phase and that of the target at 400 °C.

From the six initial experiments targeting NTMO, ARROWS$^3$ acquired knowledge regarding four different pairwise reactions (involving MoO$_2$ and various Na precursors) that led to the formation of Na$_2$Mo$_2$O$_7$ at 300 °C. To maintain a strong thermodynamic driving force to form the target, the algorithm selected all remaining experiments based on precursor sets expected to avoid pairwise reactions that formed Na$_2$Mo$_2$O$_7$ and therefore reduced ΔG′. This change in priority from ΔG (based on the precursors) to ΔG′ (based on the predicted intermediates) is highlighted by the vertical dashed line in Fig. 5a. The updated prioritization based on ΔG′ led to a clear increase in the yield of NTMO, as all experiments after the sixth iteration gave ≥30% yield of NTMO. This improvement is largely attributed to reduced Na$_2$Mo$_2$O$_7$ formation when more stable Na precursors such as Na$_2$CO$_3$ or Na$_2$TeO$_3$ are used. ARROWS$^3$ further discovered from the outcome of the 16th experiment that it is even more effective to use precursors (Na$_2$O, MoO$_3$, and TeO$_2$) that avoid Na$_2$Mo$_2$O$_7$ entirely by instead forming Na$_2$MoO$_4$. This was the only precursor set for which Na$_2$Mo$_2$O$_7$ was not detected at any temperature, and as a result, it successfully produced a sample containing 62% NTMO by weight. It did so by forming Na$_2$MoO$_4$, which retains a favorable driving force (negative reaction energy) to react with the remaining precursors and form the target:

$$Na_2MoO_4 + 3TeO_2 + 2MoO_3 \rightarrow Na_2Te_3Mo_3O_{16} (\Delta G' = -8\,meV/atom)$$

Given that the updated reaction energy is relatively small, we suspect that longer hold times could be used to improve the purity of the synthesis product. To confirm this, we prepared a new sample containing the same precursors (Na$_2$O, MoO$_3$, and TeO$_2$) and held them at the optimized synthesis temperature 400 °C for a longer hold time of 8 h. The XRD pattern of the resulting product is shown in Fig. 5b, revealing that the use of a longer hold time led to substantially improved purity. The sample contained 94% NTMO by weight, in addition to a 6% TeO$_2$ impurity. For comparison, we carried out an identical synthesis procedure using a precursor mixture where MoO$_3$ was replaced with MoO$_2$, for which the resulting product did not contain any detectable amount of NTMO (Supplementary Fig. 5). This contrasting result highlights the importance of precursor selection and its effect on the reaction pathways that proceed during synthesis. By replacing a single precursor and thus altering which intermediate phase forms first (Na$_2$Mo$_2$O$_7$ or Na$_2$MoO$_4$), the target yield can vary from 0% to >90%.

## LTOPO

As a final demonstration, ARROWS$^3$ was used to direct a series of experiments targeting the triclinic polymorph of LiTiOPO$_4$ (t-LTOPO) based on a search space consisting of 30 different precursor sets and two synthesis temperatures (400, 500, 600, 700 °C). To achieve this target, the algorithm must learn to avoid the formation of a lower-energy polymorph that exists at the same composition but adopts an orthorhombic structure (o-LTOPO)[41]. In the top panel of Fig. 6a, we plot the weight fraction obtained for each polymorph with respect to the number of precursor sets that were sampled by ARROWS$^3$ during its optimization of the synthesis process. These weight fractions are taken from experimental outcomes at 700 °C, which is the only temperature where either polymorph of LTOPO formed. The solid dots in Fig. 6a represent experimentally observed weight fractions, whereas the hollow dots represent predictions made based on the intermediates formed at 400 °C. A total of eight precursor sets were tested before identifying an optimal synthesis route for t-LTOPO, though many of these sets produced identical intermediates and therefore did not require sampling of temperatures >400 °C.

A key distinguishing feature between the precursor sets tested by ARROWS$^3$ is the amount of LiTi$_2$(PO$_4$)$_3$ formed as an intermediate in each case. The weight fraction of this phase contained in each sample

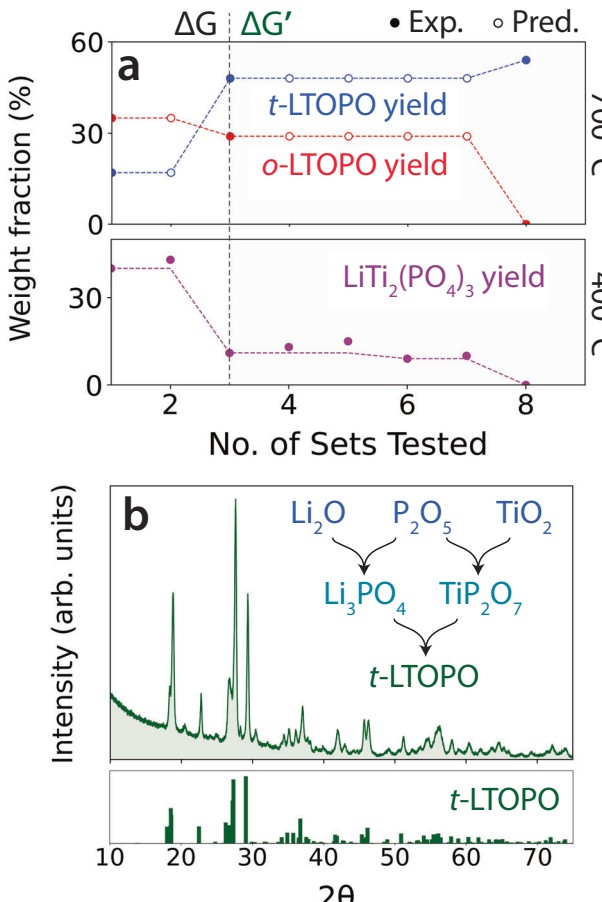

**Fig. 6 | Optimization of triclinic LiTiOPO$_4$ ($t$-LTOPO) synthesis using ARROWS[3].** **a** The top panel shows the weight fractions obtained for the target and its competing polymorph ($o$-LTOPO) based on each precursor set that was tested at 700 °C. The bottom panel displays the weight fraction of a common impurity phase, LiTi$_2$(PO$_4$)$_3$, obtained at 400 °C. Solid (hollow) dots represent experimental (predicted) values. **b** X-ray diffraction (XRD) pattern measured from the synthesis product of the optimized precursor set, Li$_2$O + TiO$_2$ + P$_2$O$_5$, which was ball milled and subsequently heated to 700 °C for 4 h. For comparison, a reference pattern for $t$-LTOPO (ICSD #39761) is also shown.

made at 400 °C is plotted in the bottom panel of Fig. 6a. Precursor sets 1–2 both formed >40% wt. of LiTi$_2$(PO$_4$)$_3$, consuming much of the driving force left to form the target. This effect is illustrated by the chemical reactions below, representing the phases contained in precursor set 1 before and after annealing at 400 °C:

$$\text{Before}: \text{LiOH} + \text{TiO}_2 + (\text{NH}_4)_2\text{HPO}_4 \rightarrow \text{LiTiOPO}_4$$
$$+ 2\,\text{NH}_3 + 2\,\text{H}_2\text{O}(\Delta G = -58\,\text{meV/atom})$$

$$\text{After}: \text{Li}_4\text{P}_2\text{O}_7 + 3\text{TiO}_2 + \text{LiTi}_2(\text{PO}_4)_3$$
$$\rightarrow 5\text{LiTiOPO}_4(\Delta G' = -6\,\text{meV/atom})$$

As outlined in recent work[50], preferential nucleation of $o$-LTOPO tends to occur when preceded by reactions with small changes in the Gibbs free energy. This is confirmed by the synthesis outcome of precursor set 1 annealed at 700 °C, which produces a sample containing 35% $o$-LTOPO and only 17% $t$-LTOPO, in addition to leftover LiTi$_2$(PO$_4$)$_3$ and TiO$_2$ impurities.

To avoid the reactions that form LiTi$_2$(PO$_4$)$_3$ and thereby retain larger $\Delta G'$ to form the target, ARROWS[3] suggests precursors where such reactions have not yet been observed. As shown by the data to the

right of the dividing line in Fig. 6a, which separates experiments selected using $\Delta G$ from those selected using $\Delta G'$, this decision successfully reduced LiTi$_2$(PO$_4$)$_3$ formation and led to increased yield of $t$-LTOPO. The plateau in the amount of each phase formed with precursor sets 3–7 is associated with the use of less reactive Li sources – including Li$_2$CO$_3$, Li$_2$TiO$_3$, and Li$_4$Ti$_5$O$_{12}$–which tend to persist until higher temperature and reduce the amount of LiTi$_2$(PO$_4$)$_3$ that forms as an intermediate. While this led to increased yield of the target, $o$-LTOPO still accompanied its formation at 700 °C. In contrast, the eighth precursor set proposed by ARROWS[3] (Li$_2$O, TiO$_2$, and P$_2$O$_5$) resulted in 54% target yield and no detectable amount of $o$-LTOPO. Notably, this was also the only precursor set that did not form any LiTi$_2$(PO$_4$)$_3$ at 400 °C. It instead formed a set of intermediates that maintained a stronger driving force to form the target as shown by the chemical reaction below:

$$\text{Li}_3\text{PO}_4 + 2\,\text{TiO}_2 + \text{TiP}_2\text{O}_7 \rightarrow 3\text{LiTiOPO}_4(\Delta G' = -24\,\text{meV/atom})$$

Because ARROWS[3] identified a synthesis route that gave 54% yield for $t$-LTOPO, exceeding our pre-defined objective of 50%, the optimization process was complete. Nevertheless, to verify that the target could made with higher purity using these optimized precursors, we separately performed a synthesis procedure where Li$_2$O, TiO$_2$, and P$_2$O$_5$ were ball milled prior to heating the mixture at 700 °C for 4 h. The XRD pattern of the resulting product is shown in Fig. 6b, revealing the formation of $t$-LTOPO without any detectable impurity phases. For comparison, the same procedure was also applied to a precursor mixture of LiOH, TiO$_2$, and P$_2$O$_5$. As shown in Supplementary Fig. 6, the resulting synthesis product contained LiTi$_2$(PO$_4$)$_3$ and $o$-LTOPO impurities, which limited the yield of $t$-LTOPO to 46% when using these non-optimized precursors.

Although $t$-LTOPO was successfully optimized, we advise careful application of ARROWS[3] for synthesizing metastable polymorphs. Our algorithm worked effectively with LTOPO, as its desired (metastable) polymorph is favored at large reaction energies, primarily due to its stable surface energy at small particle size[50]. This makes it well-suited for ARROWS[3], which learns to prioritize synthesis pathways with large reaction energy at the target-forming step. However, if the stable polymorph instead had low surface energy, its formation would be enhanced by the recommended precursor sets. Therefore, our general recommendation is to use ARROWS[3] for the following cases: (1) targets that are inherently stable; (2) targets that are metastable with respect to phase separation; and (3) targets that are metastable with respect to polymorphic transition but have lower surface energies than the ground states.

## Discussion

Precursor selection often has a marked effect on the outcomes of solid-state synthesis experiments, dictating whether they form desired products or unwanted impurities[6,7]. The importance of choosing optimal precursors is demonstrated by our syntheses targeting YBCO, for which only 10 precursor sets (out of 47 total) are successful in forming YBCO without any detectable impurity phases. Similarly, both NTMO and LTOPO were found to require the use of specific precursor sets that circumvent the formation of competing phases that otherwise limit the yield of the metastable targets. Changing just one precursor can lead to a completely different synthesis outcome, as shown by the 94% wt. increase observed in the yield of NTMO when MoO$_2$ is replaced by MoO$_3$. Understanding the origin of such large changes requires a detailed inspection of their associated reaction pathways. While this would typically be accomplished by using in situ characterization techniques, we have shown that information regarding the intermediate phases formed during solid-state synthesis can be gathered by probing different annealing temperatures with fixed hold times. For example, the low-temperature (400 °C) synthesis

experiments targeting LTOPO reveal whether $LiTi_2(PO_4)_3$ forms as an intermediate, which subsequently controls the yield of the metastable polymorph at higher temperature (700 °C).

ARROWS[3] effectively uses intermediate-phase information gleaned from low-temperature experiments to determine where a given reaction pathway goes wrong. It does so by rationalizing each set of experimental outcomes using pairwise reaction analysis, which assumes that a mixture of solid precursors reacts two phases at a time. This assumption is justified by several previous studies[6,15,51], where in situ XRD was used to verify that solid-state reactions often proceed in pairs owing to the limited diffusion lengths of species in the solid medium. In the current work, systematic pairwise reaction analysis is used to identify which precursors react to consume much of the available free energy, thereby reducing the driving force ($\Delta G'$) that remains to form the target. Once this information is known, ARROWS[3] prioritizes experiments based on precursor sets that are expected to avoid such unfavorable pairwise reactions. Our tests on the YBCO dataset showed this to be an effective approach for the rapid identification of optimal synthesis routes, as ARROWS[3] identified all ten of the best experimental procedures while sampling less than half of the entire search space. Similarly, it identified successful procedures for the synthesis of two metastable phases, NTMO and LTOPO, while requiring only 35% and 14% of their search spaces to be sampled, respectively.

Efficient data collection in vast experimental domains is a longstanding challenge. Traditional approaches based on design of experiments[52,53], including the D-optimal design algorithm tested here, can aid in the selection of experiments that are most informative to model a quantity such as target yield. However, these methods can fall short when dealing with a particularly large search space or when the quantity of interest is sparsely valued. Both challenges exist in solid-state synthesis, where many precursor combinations are often available for a given target, most of which fail to produce that target in any measurable amount. This warrants the use of active learning algorithms that can efficiently navigate the search space by adapting from failed experiments. Here we evaluated the performance of two such methods, Bayesian optimization and genetic algorithms, when applied to optimize the synthesis of YBCO. While each is known to perform well on continuous variables such as time or temperature[54,55], our tests show that they fail on the discrete task of precursor selection. We suspect their ineffectiveness is caused by using one-hot encodings to represent each precursor set, which fails to capture the similarities and differences between various chemical compounds. Recent work on organic synthesis has shown that black-box optimization techniques can perform well in the selection of molecular precursors when they are represented using physical descriptors such as SMILES strings[54]; however, no such universal representation exists for crystalline materials. Further complicating matters is the fact that precursor sets used in solid-state synthesis often have varied lengths—i.e., some sets contain more precursors than others—which make them difficult to represent using a fixed-length input vector for optimization.

ARROWS[3] systematically explores the search space associated with solid-state synthesis by actively learning from failed experiments. To overcome the limitations outlined in the previous paragraph, ARROWS[3] relies on a single metric (the remaining reaction energy) that can be updated is it reconstructs the path a given synthesis procedure takes. Previous work has demonstrated that reaction energies ($\Delta G$) often dictate the selectivity of competing phases in solid-state synthesis[6,15], and reactions with larger $\Delta G$ tend to occur more rapidly[16,37]. Initially, when no intermediates are known, the available reaction energy corresponds to the free energy difference between the target and precursors, thus motivating our choice to first prioritize experiments based on precursor sets with the largest reaction energies. Once intermediates become known, ARROWS[3] re-ranks precursor

sets based on their updated reaction energies ($\Delta G'$) remaining to form the target. Using this approach, the algorithm can discard reaction pathways that become trapped in metastable states close in energy to the target.

Notably, a unique feature of ARROWS[3] is that it becomes more efficient at identifying optimal experiments as it builds the size of its pairwise reaction database. This was demonstrated by the correlation between the frequency at which optimal synthesis routes were discovered on the YBCO dataset and the number of pairwise reactions that were collected (Fig. 3b). Further improving the utility of the pairwise reactions learned by ARROWS[3] is their transferability across materials in related chemical spaces. For example, our analysis of the YBCO experiments revealed 34 unique pairwise reactions involving common precursors for Y, Ba, and Cu. Should any of these compounds be used for the synthesis of a new material, ARROWS[3] would operate more effectively by predicting their reaction outcomes a priori. Predictions of this nature will in general become more abundant as the overlap between chemical spaces increases, specifically when considering target materials with two or more shared elements. As the decisions made by ARROWS[3] require minimal human input, the algorithm is well-suited to act as the brain behind autonomous platforms that are currently being developed[25]. With years of continuous and autonomous experimentation, such platforms could lead to the development of a standardized pairwise reaction database that covers much of the periodic table, enabling accurate predictions regarding optimal synthesis routes for new materials without requiring additional experiments. Researchers across the field of solid-state chemistry could also contribute to this database and refer to it for their own synthesis design.

There exist several opportunities to further improve the efficiency and interpretability of ARROWS[3]. The algorithm currently relies on thermodynamic arguments to optimize a target's yield, specifically by assuming that synthesis reactions with large driving force will be most effective. Future work may additionally consider the influence of kinetic factors such as diffusion and nucleation rates, though these are currently challenging to assess in a quantitative fashion due to both computational limitations and a lack of clarity on the relevant conditions under which each process should be evaluated. Related efforts have developed approximate models for nucleation rates that consider the structural similarity between precursor and target materials, in addition to their associated reaction energy[28]. Such factors could be incorporated into ARROWS[3] and its precursor ranking scheme by using structural descriptors based on matminer statistics[56] or graph-based representations[57–59]. Descriptors related to particle morphology and sample density could also be included in the optimization process, as both have been reported to affect synthesis outcomes[60,61].

Beyond the selection of optimal precursors, synthesis planning often requires the heating profile to be carefully designed. Previous work has addressed this challenge by using standard optimization techniques[55], which perform well as the heating profile can be described in terms of continuous variables (e.g., temperature and time). However, our findings show that a more physics-informed approach may also be viable. For the synthesis of each target material studied in this work, ARROWS[3] used a short hold time (4 h) to identify the precursors and temperature that give maximal target yield. If necessary, manual decisions were made to increase the hold time if (1) the target yield was lower than desired, and (2) the leftover reactions needed to grow the target were thermodynamically favorable. In doing so, >90% yield was obtained for all three target materials we considered. Moreover, it was shown that such high yield was possible only for the precursor sets optimized by ARROWS[3] at short hold time, thereby demonstrating that long hold times need not be used when testing various precursors. Decisions regarding when to extend the hold time after identifying an optimal set of precursors could later be

incorporated into ARROWS[3], enabling further progress toward complete autonomy in solid-state synthesis.

While we have shown that ARROWS[3] performs well on three benchmarks, there may still be room for improvement. To aid in the development of new algorithms for decision-making in solid-state synthesis, all data reported in this work is made publicly available. In particular, the YBCO dataset contains experimental outcomes from all the available precursor combinations. This critically includes both positive (successful) and negative (failed) synthesis outcomes, and as such, can be used to train and validate algorithms that require both types of data. We anticipate that such algorithms will not only facilitate a more systematic approach to the planning of synthesis experiments performed by human researchers, but also enable the development of fully autonomous platforms for materials development[25]. An additional benefit of ARROWS[3] specifically, when applied in conjunction with automated synthesis platforms, is that multiple successful synthesis routes can be learned for a given target. Such information on alternate experimental procedures will be valuable when more practical considerations become important, such as the optimization of morphology, synthesis cost, or the ability to industrially scale up the synthesis of a novel compound.

## Methods

### Formulation of the search space

Targeted materials synthesis can be framed as an optimization problem for which the objective is to maximize the yield of a desired phase with respect to several experimental variables including the choice of precursors, synthesis temperature, hold time, and atmospheric conditions. Here we assume a fixed hold time and set of atmospheric conditions (e.g., $p_{O_2}$ and $p_{CO_2}$) which are supplied by the user for a given target, hence constraining the search space to account only for the selection of precursors and synthesis temperature. To define this search space, ARROWS[3] requires that the user provide a list of compounds that are available to be used as precursors. From this list, all unique precursor combinations are enumerated and those that can be stoichiometrically balanced with the target are recorded as possible precursor sets for it. The number of precursors included in each set is limited to the number of elements in each target. For example, only sets containing $\leq 4$ precursors will be considered for the synthesis of a quaternary oxide containing three cations. Because carbonates, hydroxides, and high-valent oxides are often used as precursors in solid-state synthesis, ARROWS[3] accounts for the possibility of $CO_2$, $H_2O$, and $O_2$ byproducts when balancing each chemical reaction. Additional byproducts can be specified when necessary. To determine which synthesis temperatures may be tested, ARROWS[3] requires that the user supply bounds ($T_{min}$, $T_{max}$) and a sampling interval ($\Delta T$). In combination with the total number of balanced precursor sets ($N_{sets}$), this information defines the search space containing $N_{exp}$ points over which optimization is performed for a given target:

$$N_{exp} = N_{sets}\left(\frac{T_{max} - T_{min}}{\Delta T} + 1\right) \qquad (1)$$

Any prior knowledge regarding the chemical system should be used when designing the search space. For example, the lower temperature bound ($T_{min}$) may be chosen to exceed the known decomposition temperatures of all carbonates and hydroxides being considered as precursors. Similarly, the upper temperature bound ($T_{max}$) may be chosen below the melting points of the precursors if the user wishes to retain a product consisting of solid powder. With respect to precursor selection, it may often be beneficial to exclude compounds that are known to be inert in the temperature ranged being considered; however, this can also be learned by ARROWS[3] through experimentation. The algorithm's self-learning capabilities become critical in chemical systems where the precursor properties are largely unknown.

### Initial ranking by ΔG

The thermodynamic driving force behind a chemical reaction is set by the change in the Gibbs free energy (ΔG) between its products and reactants. Under constant temperature and pressure, reactions can occur spontaneously only if they reduce the Gibbs free energy (ΔG < 0) of the system. ARROWS[3] initially ranks all the available precursor sets in order of their reaction energies (ΔG) to form the target. Those with the largest (most negative) ΔG are prioritized, whereas those with ΔG ≥ 0 are excluded from consideration. For each set, ΔG of the solid compounds is determined using DFT-calculated 0 K formation energies from the Materials Project[34], along with temperature-dependent free energies approximated using the machine-learned descriptor developed by Bartel et al.[35]. In cases where a novel phase (not available in the Materials Project) is considered, we use the DFT-calculated energy of the convex hull at that phase's composition. For gaseous phases, ΔG is obtained from the experimental NIST database[62]. All reaction energies are normalized per atom of the product phase(s) formed to ensure a consistent comparison between different precursor sets.

The initial ranking by ΔG is intended to prioritize sets of precursors that are expected to react under short timescales; however, such precursors are not necessarily the most effective at forming the target. In addition to having a strong thermodynamic driving force to form the target, precursor sets with large ΔG often have similarly large driving forces to form unwanted impurity phases[7]. We have therefore designed ARROWS[3] to learn from the outcomes of failed experiments by determining which reactions led to the formation of such impurity phases. Details on this process are given in the next two sections.

Our consideration of ΔG is a simplification of the factors that dictate solid-state synthesis. In addition to selecting optimal precursors, the particle morphology and heating rate can also have a substantial influence on reaction outcomes[60,61]. Furthermore, certain compounds may react with the atmosphere prior to heating, e.g., to form carbonates or hydroxides. Such factors are currently not accounted for but could in principle be included by studying the evolution of each individual precursor as a function of temperature and time. Because this information is not generally available for all compounds and precursor powders, the current implementation of ARROWS[3] focuses only on ΔG, which is more readily calculated using the methods described in the previous few paragraphs. Future work may consider incorporating additional properties into the algorithm's ranking scheme, and further details on this possibility are provided in the Discussion section of the main text.

### Temperature selection for intermediate identification

To pinpoint the origin of any impurity phases that caused a synthesis procedure to fail, it is necessary to identify the intermediate phases that formed while heating. Previous work has demonstrated that precursors used in solid-state synthesis typically do not transform directly to the final products, but instead proceed through a series of pairwise reactions that form transient intermediate phases and incrementally reduce the free energy of the sample[6,15]. Characterizing these intermediates would traditionally require the use of in situ X-ray diffraction (XRD); however, we propose that similar information can be obtained by testing a range of synthesis temperatures for a given precursor set. Assuming that a fixed hold time is used at each temperature, the XRD patterns gathered from the resulting samples provide discrete snapshots of the reaction pathway, from which intermediate phases can be identified in a high-throughput and automated fashion using recently developed machine learning algorithms[38].

By inspecting the temperature-dependent synthesis outcomes for a given precursor set, ARROWS[3] determines which pairwise reactions occurred while heating. To this end, we assume that any phases

detected at a specific temperature ($T$) may act as reactants that lead to the formation of new phases at the next highest temperature ($T + \Delta T$). Accordingly, when XRD measurements reveal a new phase that is not present in the associated precursor set nor identified as an intermediate phase at lower temperature, ARROWS[3] is tasked with identifying the precise combination of phases responsible for its formation. If a new phase is detected at $T_{min}$, the algorithm evaluates which two-phase combination(s) from the precursor set have the appropriate compositions (i.e., can be stoichiometrically balanced) to produce that phase. In cases where there exists only one such possible combination, it is recorded as an observed pairwise reaction with an onset temperature less than $T_{min}$. A similar procedure is followed when new phases are detected at $T > T_{min}$, except that ARROWS[3] considers the intermediate phases detected at the next lowest temperature ($T - \Delta T$) as possible reactants.

Oftentimes, different sets of precursors can react to form identical sets of intermediates at low temperature, which subsequently result in the same products upon further heating[63]. To avoid testing all temperatures for such redundant synthesis routes, ARROWS[3] suggests that experiments first be performed at $T_{min}$ for each precursor set. It then checks whether the observed products and their associated weight fractions differ from those obtained using other precursors sets that were previously tested at $T_{min}$. Differences as large as 10% are allowed between two sets of products while still considering them to be identical as there is often limited precision in the refinements performed using XRD patterns from multi-phase mixtures. If the observed products for a precursor set are indeed unique, the next highest temperature ($T + \Delta T$) is proposed for that set. This process is repeated until the target is successfully obtained with sufficiently high yield, as specified by the user, or until $T_{max}$ is reached for the specified precursor set.

By default, ARROWS[3] operates under the assumption that a linear heating ramp is used to reach the specified hold temperature ($T$). In practice, however, a preheating step is occasionally used to decompose certain precursors at a temperature lower than the specified hold. For example, nitrate precursors such as $LiNO_3$ and $NaNO_3$ are often preheated to avoid rapid evolution of gases at higher temperature[64]. To handle such cases, expected decomposition temperatures and products can be incorporated into the pairwise reaction database prior to running ARROWS[3]. Without the user providing this information, the algorithm will still identify the decomposition product except in cases where that product reacts with another phase prior to XRD measurements, which would otherwise preclude its detection.

## Updated ranking by $\Delta G'$

ARROWS[3] learns from previously identified pairwise reactions to make informed decisions regarding optimal synthesis routes. It does so by predicting which intermediates will form upon heating precursor sets that have not yet been tested. An example of this process is given below for an arbitrary target ($AB_2C$):

$$\text{Precursor set not yet tested}: A + 2B + C \ (\Delta G_{initial})$$

$$\text{Previously identified pairwise reaction}: A + 2B \rightarrow AB_2 \ (\Delta G_{interm})$$

$$\text{Reaction using anticipated intermediates}: AB_2 + C \ (\Delta G' = \Delta G_{initial} - \Delta G_{interm})$$

In this example, the anticipated intermediate phases were determined based on previous synthesis outcomes that involved a reaction between $A$ and $B$. The updated reaction energy ($\Delta G'$) to form the target ($AB_2C$) is then calculated based on the intermediates ($AB_2 + C$) that result from this pairwise reaction. Similar analysis is applied to all precursor sets that have not yet been tested and their reaction energies are

updated accordingly. In cases where no intermediates can be predicted, the reaction energy remains unchanged ($\Delta G' = \Delta G$). Following these changes, precursor sets are ranked to prioritize reactions with the most negative $\Delta G'$, i.e., those with the largest thermodynamic driving force at the presumed target-forming step. ARROWS[3] uses the updated ranking to continually suggest new precursor sets until an experiment is found that gives sufficiently high yield of the target phase (as specified by the user) or until all precursor sets have been tested.

We acknowledge that it is generally difficult to ascertain whether a given reaction energy is large enough for the associated transformation to occur within a reasonably short timeframe. The reaction rate is determined not only by the energy change, but also by several factors related to diffusion and nucleation These rates are highly non-trivial to predict and strongly dependent on the specific chemistry being considered. Given these considerations, ARROWS[3] is designed to rank various precursor sets based on their relative reaction energies to form a target phase, whether from the initial precursors ($\Delta G$) or from the intermediates that form during synthesis ($\Delta G'$). While this affects the order in which different precursor sets are tested, none are excluded for having a low reaction energy. Instead, such precursors will be tested at a later stage in the optimization process, if necessary.

## YBCO synthesis

The synthesis of YBCO is most commonly performed using $Y_2O_3$, $CuO$, and $BaCO_3$[42]. This combination of precursors requires >12 h of annealing at 950 °C, in addition to intermittent regrinding, to eliminate the unwanted impurity phases that often appear. In contrast, recent work has shown that by replacing $BaCO_3$ with $BaO_2$, YBCO can be obtained with high purity while using a shorter anneal time of 30 min[6,43]. These findings highlight the importance of precursor selection and its effect on the yield of YBCO under limited hold time, making it a suitable test case for ARROWS[3]. To this end, we considered 11 common precursors from the Y–Ba–Cu–O space: $Y_2O_3$, $Y_2(CO_3)_3$, $BaO$, $BaCO_3$, $BaO_2$, $CuO$, $CuCO_3$, $Cu_2O$, $BaCuO_2$, $Ba_2Cu_3O_6$, and $Y_2Cu_2O_5$. These compounds were combined to form 47 different precursor sets, listed in Supplementary Table 1, that were each tested at four synthesis temperatures (600, 700, 800, and 900 °C) using a fixed hold time of 4 h.

All binary phases listed in the above paragraph (including the carbonates) were purchased from Sigma-Aldrich, whereas the ternaries ($Y_2Cu_2O_5$, $BaCuO_2$, and $Ba_2Cu_3O_6$) were synthesized in-house. For each ternary phase, stoichiometric amounts of the starting materials were mixed in ethanol with six 10-mm stainless steel balls using a high-energy SPEX mill (SPEX SamplePrep 8000 M) for 9 min. The resulting slurry was dispensed into a crucible and dried at 80 °C, followed by a high-temperature anneal at the specified synthesis temperature for each sample. $Y_2Cu_2O_5$ was made from $Y_2O_3$ and $CuCO_3$ using a 12 h hold at 1050 °C. $BaCuO_2$ was synthesized from $BaCO_3$ and $CuO$ using a 24 h hold at 910 °C. $Ba_2Cu_3O_6$ was prepared from $BaO_2$ and $CuO$ using a 24 h hold at 600 °C. The corresponding XRD patterns, shown in Supplementary Fig. 7, point to successful synthesis outcomes as each sample contains the desired ternary phase with minimal impurities.

To assess the phase purity for each synthesis product, XRD measurements were performed with an Aeris diffractometer from Panalytical. We used XRD-AutoAnalyzer[38] to analyze the resulting XRD patterns and identify any crystalline phases present. This algorithm relies on a convolutional neural network to map each pattern onto a set of constituent phases. Here we trained the network on all phases reported in the ICSD within the space of Y–Ba–Cu–O chemistries. After identifying the phases in each pattern using XRD-AutoAnalyzer, their weight fractions were evaluated through analysis of relative peak intensities. A more careful approach based on Rietveld refinement, which accounts for properties such as grain size and texture, would be required to obtain precise weight fractions. However, this work only requires that we compare relative weight fractions between different experiments, enabling ARROWS[3] to identify the most effective

synthesis route for a given target. For YBCO specifically, all experiments were performed prior to optimization, and therefore it was used to evaluate the performance of several algorithms including ARROWS[3], Bayesian optimization, genetic algorithms, and D-optima design.

## NTMO synthesis

The initial discovery of NTMO was enabled by the use of a hydrothermal synthesis procedure whereby an aqueous solution of $Na_2TeO_3$, $TeO_2$, and $MoO_3$ was held at 220 °C for 48 h[40]. More recently, a solid-state synthesis route was also reported: $Na_2CO_3$, $TeO_2$, and $MoO_3$ were mixed and held at 430 °C for 48 h with intermittent regrinding[49]. We suspect that ARROWS[3] can handle the synthesis of phases such as NTMO, which are metastable with respect to decomposition, as it should learn to avoid the formation of any competing phases that result in an unfavorable driving force ($\Delta G > 0$) to form the target. For the experimental campaign targeting NTMO, eleven precursors were purchased from Sigma-Aldrich: $Na_2O$, $Na_2CO_3$, $NaOH$, $Na_2O_2$, $MoO_2$, $MoO_3$, $TeO_2$, $Na_2TeO_3$, $Na_2MoO_4$, $Na_2Mo_2O_7$, and $(NH_4)_2MoO_4$. A total of 23 precursor sets were considered (Supplementary Table 2), for which synthesis temperatures of 300 and 400 °C were tested at a fixed hold time of 4 h. We avoided the use of higher temperatures as melting is expected to occur near 450 °C, making the product difficult to extract[40]. In contrast to the YBCO campaign, where all possible experiments were performed and ARROWS[3] was only applied post hoc, the LTOPO experiments were carried out under the guidance of ARROWS[3] until NTMO was obtained with a weight fraction exceeding 50%. No black-box optimization techniques were used to explore this dataset as only part of the design space was sampled by ARROWS[3].

## LTOPO synthesis

The tendency for LTOPO to crystallize in its triclinic polymorph, as opposed to its orthorhombic ground state, is highly sensitive to the choice of precursors and synthesis temperature[41,65]. Recent work has proposed that the t-LTOPO nucleates first owing to its more stable surface energy, which dictates the relative nucleation rate of each polymorph when $\Delta G$ is large[50]. Therefore, although ARROWS[3] encodes no structural information and is not designed for the synthesis of metastable polymorphs in general, we believe it is well-suited for t-LTOPO (and similarly stabilized metastable polymorphs) since it aims to identify reaction pathways that maintain large $\Delta G$. Ten commercially available phases were purchased from Sigma-Aldrich and used as precursors: $Li_2O$, $Li_2CO_3$, $LiOH$, $TiO_2$, $P_2O_5$, $NH_4H_2PO_4$, $(NH_4)_2HPO_4$, $Li_3PO_4$, $Li_2TiO_3$, and $Li_4Ti_5O_{12}$. A total of 30 precursor sets, listed in Supplementary Table 3, were considered. Four synthesis temperatures (400, 500, 600, and 700 °C) were sampled for each set at fixed a hold time of 4 h. Synthesis experiments were performed under the guidance of ARROWS[3] until t-LTOPO was obtained with a weight fraction exceeding 50%. No black-box optimization techniques were applied.

### Reporting summary

Further information on research design is available in the Nature Portfolio Reporting Summary linked to this article.

## Data availability

The phase information associated with the experimental outcomes in the YBCO, NTMO, and LTOPO synthesis datasets can be found in the ARROWS[3] repository. Source data for all graphs are also provided as a Source Data file with this publication. Source data are provided with this paper.

## Code availability

ARROWS[3] can be accessed in the public repository, https://github.com/njszym/ARROWS, which is archived at https://doi.org/10.5281/zenodo.8331001.

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

## Acknowledgements

This work was intellectually led by the D2S2 program within the U.S. Department of Energy, Office of Basic Energy Sciences, Materials Sciences and Engineering Division under Contract No. DE-AC02-05-CH11231. The experiments were performed with support from the Laboratory Directed Research and Development Program of Lawrence Berkeley National Laboratory. Additional support was provided by Umicore Specialty Oxides and Chemicals. Computations were performed using the National Energy Research Scientific Computing Center (NERSC), which is supported by the Office of Science and the U.S. Department of Energy under Contract No. DE-AC02-05CH11231. N.J.S.

was supported in part by the National Science Foundation Graduate Research Fellowship under grant #1752814.

## Author contributions

N.J.S. developed the algorithm for automated precursor selection and prepared the manuscript. P.N. performed the synthesis experiments, characterized the products, and interpreted the resulting X-ray diffraction patterns. Y.Z. contributed to experimental design and analysis. The project was conceived by N.J.S. and C.J.B., and its execution was supervised by G.C. and Y.Z. All authors reviewed and edited the final manuscript.

## Competing interests

The authors declare no competing interests.
