## [Peer Review File · Nature Communications]

REVIEWER COMMENTS

Reviewer #1 (Remarks to the Author):

In their paper, Symanski et al. describe a system by which inorganic syntheses can be systematically improved and understood through an iterative procedure. In general, I think this paper is excellent and deserves to be published, ideally soon given how quickly this topic in the community. I also believe the paper needs very few additions, as it contains a lot of valuable information, a very-well informed commentary, and a nice clarification of the limits and practical considerations of the software tool (which is far too uncommon in modern methods papers).

I only have one recommendation for the paper, which is to make the discussion of structural and compositional features a little more thorough. My understanding of ARROWS from this paper is that the only information that is used a-priori about precursor structures and formulas are derived from DFT reaction energies. Honestly, I appreciate this - I think it makes the system more interpretable and physical, but presumably precursor structure does matter and could be used predictively. The authors claim that there is no universal structural representation, which is true, but there are at least two options (standard matminer site-statistic and graph-based) that are very commonly used today and would be reasonable to try. The authors don't need to do this (at least not for this paper), but I would comment on how structural features might be used in either future iterations of the tool or in other tools to understand how precursor structure relates to reactivity and selectivity of synthesis.

Reviewer #2 (Remarks to the Author):

Review overview

- This paper aims to optimize the yield of the target phase in solid-state reactions, proposing a method to select precursors (starting material or intermediates) with a large negative energy difference from the target phase using first-principles calculations. Furthermore, the method aims to improve the efficiency of yield enhancement of the target phase by feeding back actual synthesis experiment results, adding precursors not on the initial candidate list, and newly considering the presence of inert intermediates.
- Reaction process design in reactions between inorganic solids has yet to be established due to the complexity of long-range atomic diffusion and its reactive processes. However, it's a cutting-edge theme approached by several groups, including the authors, leveraging first-principles calculations and machine learning techniques.
- Attempts to evaluate the driving force of reactions from first-principles calculation results based on thermodynamic backgrounds have already been performed several times, as cited in this paper, with some success for various compounds. What makes this method unique is the addition of feedback from unsuccessful synthesis results.
- The authors have successfully improved yields with what seems to be a relatively small number of trials by applying this method to three target phases as case studies. The selection of the three

target phases and the synthesis conditions are appropriate for demonstrating this method, revealing the authors' profound expertise.

- This paper succinctly summarizes the authors' ideas and actual processes that underpin this method and can evaluate performance based on appropriate results.

- On the other hand, from the viewpoint of a synthesis researcher, I don't feel any impact from this method. To make the paper more understandable to synthesis researchers, you might find the following comments helpful.

Comments

- As seen with the three sample cases presented this time, when the desired compounds are known, approximately 200 solid-state synthesis reactions are not particularly difficult for synthesis researchers. The proposed method here suggests a reduction in experimental repetitions by iteratively and sequentially selecting precursors and conducting synthesis experiments. However, performing possible combinations simultaneously and in parallel is a conventional method in synthesis processes. Even if parallel experimentation leads to a larger number of trials, it might eventually lead to the discovery of better synthesis conditions in a shorter time.

- As the authors mentioned in the method section, this approach is only applicable when the crystal structures of the target phase and precursors are clearly known, as it uses first-principles calculations. For instance, when the target phase has partially occupied sites or is a completely new material, or when the starting materials are amorphous, or when hydroxides, acid hydroxides, or hydrates with unclear hydrogen sites are used, evaluating their energy is challenging. Also, in solid-state synthesis, the reactivity changes even with the same starting material if the particle size or crystallinity changes, and the time required for the reaction also changes with the density of the compacted powder. A preliminary calcination process, which holds at a slightly lower temperature than the actual calcination for a short time, can also effectively contribute to single-phase formation. However, it seems such factors are not explicitly considered. It may be helpful to separate the method into the calculation and experimental sections and provide more specific examples of its application range in each.

- This paper lacks citations of important methods concerning synthesis process design in solid-state reactions. Several are introduced in the second paragraph on page 3, but the only method using machine learning introduced is from the authors' own group. It does not cite studies that predict synthesis results using a recommender system and parallel synthesis experiments (for example, [10.1021/acs.chemmater.9b01799](https://doi.org/10.1021/acs.chemmater.9b01799)). I believe that with the utilization of robotics, machine learning methods based on in-house experimental data will become increasingly important, even though their examples were limited due to the limited number of synthesis experiments.

- In the YBCO example, the authors mention the well-known reason for the low yield when BaCO₃ is used as a starting material, which is due to its high decomposition temperature. If a human researcher were to assume the current calcination temperature and hold time, BaCO₃ would not be used from the start, and thus, this method seems to be doing something unnecessary. Similarly, in the synthesis of NTMO, the inclusion of MoO₂, which is known to have a melting point several hundred Kelvin higher than MoO₃, as a precursor candidate in low-temperature synthesis would not be conceivable from a synthesis researcher's perspective. Can't this kind of thing be incorporated into the method as prior knowledge?

- The evaluation that BaCuO₂ and Y₂BaCuO₅ are inert and thus planning the experiment to exclude them from the synthesis path is not sufficiently explained. For instance, the fact that their energy is close to YBCO and consumes much of the driving force of the starting material is not a valid reason. This is because, as the authors show in the NTMO example, a reaction can occur even with a small driving force. In particular, in the YBCO case, since the calcination time is limited, whether or not

they are inert seems to depend on prior knowledge from previous research. Therefore, it does not seem to me to be the "Autonomous decision making" mentioned in the title.

- Does the set of 47 precursors in the YBCO case cover all combinations that could potentially produce YBCO in stoichiometric proportions? In relation to this, in Figure 4, white squares seem to represent untested combinations, but were they determined to be unnecessary for exploration using the current method even after 188 experiments?
- In the YBCO case, why is the number of syntheses necessary to find all 10 good yield conditions used as the standard for comparison between BO, GA, and the current method? From the viewpoint of improving the synthesis process, it should be sufficient if any one of the ten can be found in the shortest possible time. Does the current method still show significant superiority when evaluated with such a metric?
- As the authors mention, the way information is fed to the predictive model in BO or GA seems unfair compared to the current method, and I did not feel the comparison itself was meaningful. Rather, I believe it is necessary to compare with experimental design methods such as the D-optimal design.
- In the case of NTMO, there is a description that extending the calcination time improves the yield because $\Delta G'$ is small. Is this decision made automatically by the current method? If not, the "Autonomous decision making" in the paper title feels exaggerated.
- The chemical reaction formulas annotated on the XRD profile in Figure 1b need an explanation as to whether they indicate a combination of precursors or phases identified from XRD.
- The temperature is inconsistent between the caption of Figure 5a and the main text.

Reviewer #3 (Remarks to the Author):

The focus of this manuscript is the development of an autonomous decision making approach for inorganic solid state syntheses. The use of high temperature solid state syntheses has, of course, results in a host of technologically important materials. While the mechanical steps of this synthetic approach appear simple, a great deal of complexity lies under the surface. The authors nicely focus on an often unobserved, but critically important component of these reactions, the formation of intermediate phase that either enable or preclude the formation of the desired product.

1. A central question that remains in the mind of this reviewer is the construction of the initial reaction parameters. These include, temperature ranges, hold times, the presence or absence of regrinding steps, and reaction conditions (oxidizing, reducing, inert atmospheres).

- The authors note that hold times are specified for each study. The chosen times are rather short (4 or 8 h) with respect to more tradition high temperature solid state approaches. How were these times chosen? The choices of reaction time between regrinding steps more often conform to experimenter schedules than the reactions themselves (there is nothing magical about 24 h), and shorter times are desired of course. Having said that, there is some concern that short times without repeated regrinding steps selects for kinetic products in reactions generally governed by thermodynamics.

- The chemistry of many solid-state reactions can and does vary as a function of temperature. By this I mean that the phase diagram being explored does not remain unchanged as the temperature increases or decreases. For example, the authors correctly note that the traditional synthesis temperature for YBCO lies above the temperature window explored in this work. The identification

of BaCO₃ as a problematic reactant at lower temperatures does not map onto its demonstrated utility at 950 C.

2. A second central theme is a quest routes for phase pure samples of known or unknown materials as quickly as possible. The infrastructure balance has shifted in this proposed synthetic methodology from traditional approaches. Traditionally one would run many reactions in parallel in a single furnace, as composition can vary between samples but temperature cannot. Conducting a range of reagent combinations at different temperatures either requires more furnaces or more sequential studies. The authors should comment on this balance.

3. The possibility of transfer learning in this approach is interesting and enticing. The authors note the possibility of predicting the formation of other compounds within the Y-Ba-Cu-O system (or more accurately – compound that could be made from the reagents used in the YBCO study). Does transfer learning extend between systems? If, for example, one was to explore BSCCO, would the YBCO work transfer?

4. The identification of pair wise interactions that result in either reactive or inert intermediates is both important and informative.

5. The authors should be cautious in the determination of weight percent using diffraction data. Many parameters affect peak intensity (or area) that are distinct from weight percent. These include grain size, crystallinity, average atomic scattering factor to name three.

Reviewer #1

General Assessment:

In their paper, Szymanski *et al.* describe a system by which inorganic syntheses can be systematically improved and understood through an iterative procedure. In general, I think this paper is excellent and deserves to be published, ideally soon given how quickly this topic develops in the community. I also believe the paper needs very few additions, as it contains a lot of valuable information, a very-well informed commentary, and a nice clarification of the limits and practical considerations of the software tool (which is far too uncommon in modern methods papers).

I only have one recommendation for the paper, which is to make the discussion of structural and compositional features a little more thorough. My understanding of ARROWS from this paper is that the only information that is used a-priori about precursor structures and formulas are derived from DFT reaction energies. Honestly, I appreciate this - I think it makes the system more interpretable and physical, but presumably precursor structure does matter and could be used predictively. The authors claim that there is no universal structural representation, which is true, but there are at least two options (standard matminer site-statistic and graph-based) that are very commonly used today and would be reasonable to try. The authors don't need to do this (at least not for this paper), but I would comment on how structural features might be used in either future iterations of the tool or in other tools to understand how precursor structure relates to reactivity and selectivity of synthesis.

Response:

We thank the reviewer for their positive comments. The possibility of incorporating structure-based descriptors into the workflow for synthesis design and decision-making is an interesting point that indeed warrants further consideration. We have now added a paragraph to the **Discussion** section of the paper, describing the potential ways in which such information may be utilized in future iterations of our algorithm.

Discussion, Page 23: There exist several opportunities to improve the efficiency and interpretability of ARROWS³. The algorithm currently relies on thermodynamic arguments to optimize a target's yield, specifically by assuming that synthesis reactions with large driving force will be most effective. Future work may additionally consider the influence of kinetic factors such as diffusion and nucleation rates, though these are currently challenging to assess in a quantitative fashion due to both computational

limitations and a lack of clarity on the relevant conditions under which each process should be evaluated. Related efforts have developed approximate models for nucleation rates that consider the structural similarity between precursor and target materials, in addition to their associated reaction energy²⁸. Such factors could be incorporated into ARROWS³ and its precursor ranking scheme by using structural descriptors based on matminer statistics⁵⁶ or graph-based representations⁵⁷⁻⁵⁹. Descriptors related to particle morphology and sample density could also be included in the optimization process, as both have been reported to affect synthesis outcomes^{60,61}.

Reviewer #2

General Assessment:

This paper aims to optimize the yield of the target phase in solid-state reactions, proposing a method to select precursors (starting material or intermediates) with a large negative energy difference from the target phase using first-principles calculations. Furthermore, the method aims to improve the efficiency of yield enhancement of the target phase by feeding back actual synthesis experiment results, adding precursors not on the initial candidate list, and newly considering the presence of inert intermediates. Reaction process design in reactions between inorganic solids has yet to be established due to the complexity of long-range atomic diffusion and its reactive processes. However, it's a cutting-edge theme approached by several groups, including the authors, leveraging first-principles calculations and machine learning techniques. Attempts to evaluate the driving force of reactions from first-principles calculation results based on thermodynamic backgrounds have already been performed several times, as cited in this paper, with some success for various compounds. What makes this method unique is the addition of feedback from unsuccessful synthesis results. The authors have successfully improved yields with what seems to be a relatively small number of trials by applying this method to three target phases as case studies. The selection of the three target phases and the synthesis conditions are appropriate for demonstrating this method, revealing the authors' profound expertise. This paper succinctly summarizes the authors' ideas and actual processes that underpin this method and can evaluate performance based on appropriate results. On the other hand, from the viewpoint of a synthesis researcher, I don't feel any impact from this method. To make the paper more understandable to synthesis researchers, you might find the following comments helpful.

Response:

We thank the reviewer for their feedback. Each comment is addressed in detail below.

Comment 1:

As seen with the three sample cases presented this time, when the desired compounds are known, approximately 200 solid-state synthesis reactions are not particularly difficult for synthesis researchers. The proposed method here suggests a reduction in experimental repetitions by iteratively and sequentially selecting precursors and conducting synthesis experiments. However, performing possible combinations simultaneously and in parallel is a conventional method in synthesis processes. Even if parallel

experimentation leads to a larger number of trials, it might eventually lead to the discovery of better synthesis conditions in a shorter time.

Response:

The reviewer raises a good point regarding the use of parallelization in synthesis experiments. While our method is generally designed to handle sequential experiments so that one synthesis outcome can be used to inform the next choice of experimental parameters, it can also be applied with batched experiments performed in parallel. To this end, the user may specify a *batch size* when running ARROWS³, which will control how many experiments are suggested at once. The algorithm will then learn from the outcomes of these experiments simultaneously before suggesting the next batch of samples to evaluate.

To showcase the influence of parallelization on the efficiency of ARROWS³, we have now performed additional tests on the YBCO dataset. Four optimization campaigns were carried out, each with a different batch size, and the resulting curves are plotted in **Supplementary Fig. 4a** (see below).

Supplementary Figure 4: (a) Number of optimal synthesis routes for YBCO identified with respect to the number of experimental samples queried by ARROWS³. Each curve represents a single optimization campaign performed using a different batch size.

The results demonstrate that the optimization process becomes less efficient when a larger batch size is used, requiring more samples to be tested before all optimal synthesis routes can be found. However, as the reviewer points out, the use of larger batches may still be beneficial when parallelization is available, and the user wishes to exhaust the search space as quickly as possible. Indeed, substantially fewer experimental iterations (*i.e.*, batches) are required to identify all optimal synthesis routes when a larger batch size is used.

To better illustrate the tradeoff that exists between the number of samples and batches required to complete the optimization process, we have created **Supplementary Fig. 4b** (see below).

Supplementary Figure 4: (b) The number of batches and individual samples required to identify all ten optimal synthesis routes. Each dot represents the requirements for one optimization campaign performed with a distinct batch size.

This plot reveals a clear anti-correlation between the number of samples and batches required to identify all optimal synthesis routes for YBCO. Whether fewer samples or batches are preferred is up to the user, who may balance the tradeoff between these two quantities by setting the batch size accordingly.

These results are now discussed in the manuscript, and the relevant text copied below.

Results, YBCO, Pages 14-15: The results presented in **Fig. 3** and **Fig. 4** were obtained by querying experiments in a serial (one-by-one) fashion. This allowed ARROWS³ to continually learn from each experimental outcome and update its ranking of precursor sets accordingly. However, traditional experiments are often parallelized. For example, multiple sets of precursors with a shared synthesis temperature may be tested simultaneously by loading them into one furnace³³. Such an approach is also compatible with ARROWS³, for which a batch size can be specified to control how many experiments are suggested at each iteration. As shown in **Supplementary Fig. 4**, the use of a larger batch size reduces the number of iterations (*i.e.*, batches) required to identify all the optimal synthesis routes for YBCO. However, because a larger batch size limits the opportunities where ARROWS³ can learn and update its ranking, it also leads to a larger number of individual samples that must be queried to identify the optimal routes. Hence, there exists a tradeoff between the number of batches and individual samples required to complete the optimization process, and the batch size acts as a hyperparameter to adjust this tradeoff depending on the user's objectives and experimental setup.

Comment 2:

As the authors mentioned in the method section, this approach is only applicable when the crystal structures of the target phase and precursors are clearly known, as it uses first-principles calculations. For instance, when the target phase has partially occupied sites or is a completely new material, or when the starting materials are amorphous, or when hydroxides, acid hydroxides, or hydrates with unclear hydrogen sites are used, evaluating their energy is challenging. Also, in solid-state synthesis, the reactivity changes even with the same starting material if the particle size or crystallinity changes, and the time required for the reaction also changes with the density of the compacted powder. A preliminary calcination process, which holds at a slightly lower temperature than the actual calcination for a short time, can also effectively contribute to single-phase formation. However, it seems such factors are not explicitly considered. It may be helpful to separate the method into the calculation and experimental sections and provide more specific examples of its application range in each.

Response:

We agree with many of these points and have added text throughout the **Methods** section to better explain the limitations of ARROWS³, as well as to outline which aspects can be customized by the user. Furthermore, we have added a new paragraph to the **Discussion** section outlining possible improvements that could be made to the algorithm. All related text is copied below.

Methods, Initial ranking by ΔG , Page 26: In cases where a novel phase (not available in the Materials Project) is considered, we use the DFT-calculated energy of the convex hull at that phase's composition.

Methods, Initial ranking by ΔG , Page 27: Our consideration of ΔG is a simplification of the factors that dictate solid-state synthesis. In addition to selecting optimal precursors, the particle morphology and heating rate can also have a substantial influence on reaction outcomes^{60,61}. Furthermore, certain compounds may react with the atmosphere prior to heating, *e.g.*, to form carbonates or hydroxides. Such factors are currently not accounted for but could in principle be included by studying the evolution of each individual precursor as a function of temperature and time. Because this information is not generally available for all compounds and precursor powders, the current implementation of ARROWS³ focuses only on ΔG , which is more readily calculated using the methods described in the previous few paragraphs.

Methods, Temperature selection for intermediate identification, Pages 28-29: By default, ARROWS³ operates under the assumption that a linear heating ramp is used to reach the specified hold temperature (T). In practice, however, a preheating step is occasionally used to decompose certain precursors at a temperature lower than the specified hold. For example, nitrate precursors such as LiNO_3 and NaNO_3 are often preheated to avoid rapid evolution of gases at higher temperature⁶⁴. To handle such cases, expected decomposition temperatures and products can be incorporated into the pairwise reaction database prior to running ARROWS³. Without the user providing this information, the algorithm will still identify the decomposition product except in cases where that product reacts with another phase prior to XRD measurements, which would otherwise preclude its detection.

Discussion, Page 23: There exist several opportunities to improve the efficiency and interpretability of ARROWS³. The algorithm currently relies on thermodynamic arguments to optimize a target's yield, specifically by assuming that synthesis reactions with large driving force will be most effective. Future work may additionally consider the influence of kinetic factors such as diffusion and nucleation rates, though these are currently challenging to assess in a quantitative fashion due to both computational limitations and a lack of clarity on the relevant conditions under which each process should be evaluated. Related efforts have developed approximate models for nucleation rates that consider the structural similarity between precursor and target materials, in addition to their associated reaction energy²⁸. Such factors could be incorporated into ARROWS³ and its precursor ranking scheme by using structural

descriptors based on matminer statistics⁵⁶ or graph-based representations^{57–59}. Descriptors related to particle morphology and sample density could also be included in the optimization process, as both have been reported to affect synthesis outcomes^{60,61}.

Comment 3:

This paper lacks citations of important methods concerning synthesis process design in solid-state reactions. Several are introduced in the second paragraph on page 3, but the only method using machine learning introduced is from the authors' own group. It does not cite studies that predict synthesis results using a recommender system and parallel synthesis experiments (*e.g.*, 10.1021/acs.chemmater.9b01799). I believe that with the utilization of robotics, machine learning methods based on in-house experimental data will become increasingly important, even though their examples were limited due to the limited number of synthesis experiments.

Response:

The introduction has now been updated to include further discussion regarding previous work on the optimization of synthesis procedures in materials science and chemistry. Relevant citations, including the work highlighted by the reviewer, have also been added. Please find the revised text below.

Introduction, Page 3: In the place of fixed ranking schemes, active learning algorithms have been used for the optimization of synthesis procedures^{29,30}. These algorithms can adapt from failed experiments and decide which parameters should be tested in later iterations. Bayesian optimization and genetic algorithms have each found success when coupled with synthesis techniques based on flow chemistry³¹ and thin film deposition³². However, these “black box” approaches are often restricted to handling continuous variables such as temperature and time, while categorical variables are more difficult to optimize. For example, choosing which precursors to use for the synthesis of a novel material is particularly challenging as it involves discrete selections from a vast range of chemical compositions and structures, instead of simply fine-tuning parameters on a continuous scale. Recent work has made progress on this front by combining parallel synthesis experiments with tensor decomposition analysis, which can be used to predict the most effective starting materials and processing conditions from just a subset of their possible combinations³³.

Comment 4:

In the YBCO example, the authors mention the well-known reason for the low yield when BaCO₃ is used as a starting material, which is due to its high decomposition temperature. If a human researcher were to assume the current calcination temperature and hold time, BaCO₃ would not be used from the start, and thus, this method seems to be doing something unnecessary. Similarly, in the synthesis of NTMO, the inclusion of MoO₂, which is known to have a melting point several hundred Kelvin higher than MoO₃, as a precursor candidate in low-temperature synthesis would not be conceivable from a synthesis researcher's perspective. Can't this kind of thing be incorporated into the method as prior knowledge?

Response:

Information regarding the utility of certain precursors and temperatures can indeed be used to aid in the algorithm's decision making. In its current implementation, such prior knowledge should be referred to when designing the search space over which ARROWS³ performs its optimization. This is now discussed in detail in the **Methods** section, with the associated text copied below.

Methods, Formulation of the search space, Page 26: Any prior knowledge regarding the chemical system should be used when designing the search space. For example, the lower temperature bound (T_{\min}) may be chosen to exceed the known decomposition temperatures of all carbonates and hydroxides being considered as precursors. Similarly, the upper temperature bound (T_{\max}) may be chosen below the melting points of the precursors if the user wishes to retain a product consisting of solid powder. With respect to precursor selection, it may often be beneficial to exclude compounds that are known to be inert in the temperature range being considered; however, this can also be learned by ARROWS³ through experimentation (see **YBCO** in the main text). The algorithm's self-learning capabilities become critical in chemical systems where the precursor properties are largely unknown.

However, for the examples provided in our work, we used minimal prior information when designing the search spaces. This was done to mimic the exploration of novel chemistries where little is known about the system beforehand, making them particularly challenging for ARROWS³ to deal with.

We do agree that the optimization could be accelerated if the available precursors were chosen more intelligently, as the reviewer suggests. To demonstrate this, we performed a new optimization campaign

targeting YBCO, this time excluding BaCO_3 as it is known to be relatively inert below its decomposition temperature. The resulting optimization curve is shown in **Supplementary Fig. 1** (see below).

Supplementary Figure 1: Number of optimal synthesis routes for YBCO identified as a function of the experimental iterations required by ARROWS³. The blue line represents optimization performed throughout the entire search space (see **Methods** in the main text), while the red line represents optimization performed in that space while excluding BaCO_3 as a precursor. This test is designed to probe the effect of incorporating prior knowledge into the search space, as a domain expert may decide to exclude BaCO_3 owing to its high decomposition temperature.

As anticipated, all ten optimal synthesis routes were identified by ARROWS³ while requiring fewer experimental iterations. These results are now discussed in the main text. Please find the additions below.

Results, YBCO, Pages 8-9: The fourth most common impurity is BaCO_3 , which is likely slow to react owing to its high decomposition temperature in air (1000 °C)^{44,45}. We note that such information could in principle be leveraged when designing the search space, *e.g.*, by removing BaCO_3 from the list of precursors since the proposed temperature range lies below its known decomposition temperature. Indeed, doing so reduces the number of experiments required to identify all optimal synthesis routes from 87 to 70 (**Supplementary Fig. 1**).

Comment 5:

The evaluation that BaCuO₂ and Y₂BaCuO₅ are inert and thus planning the experiment to exclude them from the synthesis path is not sufficiently explained. For instance, the fact that their energy is close to YBCO and consumes much of the driving force of the starting material is not a valid reason. This is because, as the authors show in the NTMO example, a reaction can occur even with a small driving force. In particular, in the YBCO case, since the calcination time is limited, whether or not they are inert seems to depend on prior knowledge from previous research. Therefore, it does not seem to me to be the "Autonomous decision making" mentioned in the title.

Response:

We agree that even reactions with small driving force can sometimes proceed on a reasonably short timeframe. However, whether this is true depends on the kinetics of the processes involving the materials at hand. A fixed reaction energy may be considered large or small depending on the chemical space. For example, the precursor sets available for NTMO span a relatively narrow range of reaction energies ($\Delta G > -279$ meV/atom) when compared with those for YBCO ($\Delta G > -684$ meV/atom). As such, a small reaction energy in the YBCO space might be considered much larger in the NTMO space.

A key principle of ARROWS³ is that it ranks precursor sets by their *relative* reaction energies within a given chemical space. This operates under the assumption that when chemistry is fixed, reactions with larger driving forces will generally occur more rapidly than those with smaller driving forces.

We would also like to stress that a precursor set will never be *excluded* for having a low reaction energy. Instead, it will simply be given lower priority in the ranking formed by ARROWS³. The algorithm skips a precursor set only when it can predict that set evolves to form a set of intermediates that were previously observed to be unsuccessful. This is detailed in the third paragraph of the **Methods** section, within **Temperature selection for intermediate identification**.

To clarify the role of reaction energy in the ranking of precursor sets, a new paragraph (copied below) has been added to the **Methods** section.

Methods, Updated ranking by $\Delta G'$, Pages 29-30: We acknowledge that it is generally difficult to ascertain whether a given reaction energy is large enough for the associated transformation to occur

within a reasonably short timeframe. The reaction rate is determined not only by the energy change, but also by several factors related to diffusion and nucleation. These rates are highly non-trivial to predict and strongly dependent on the specific chemistry being considered. Given these considerations, ARROWS³ is designed to rank various precursor sets based on their *relative* reaction energies to form a target phase, whether from the initial precursors (ΔG) or from the intermediates that form during synthesis ($\Delta G'$). While this affects the order in which different precursor sets are tested, none are excluded for having a low reaction energy. Instead, such precursors will be tested at a later stage in the optimization process, if necessary.

Comment 6:

Does the set of 47 precursors in the YBCO case cover all combinations that could potentially produce YBCO in stoichiometric proportions? In relation to this, in Figure 4, white squares seem to represent untested combinations, but were they determined to be unnecessary for exploration using the current method even after 188 experiments?

Response:

Yes, the 47 precursor sets evaluated in our experiments do cover all possible combinations whose stoichiometric proportions are uniquely defined with respect to YBCO. In **Fig. 4**, the white squares do not represent pairs of reactants that were untested; rather, they represent pairs whose reactivity could not be determined with a high degree of confidence. There are two causes for this, each of which are now discussed in the **Results** section of the manuscript (revised text copied below).

Results, YBCO, Page 13: There also exist some pairs of compounds whose reactivity was not learned by ARROWS³ during its optimization of YBCO synthesis. These 23 pairs are denoted by the white (unshaded) squares in **Fig. 3**. We observe two factors that prevent ARROWS³ from learning pairwise reaction information. First, when two phases (*e.g.*, A|B) react in a three-phase set (A, B, and C), the algorithm is unable to learn how the remaining phase (C) interacts with the already reacted compounds (A and B). Separate experiments based on the individual pairs (A|C and B|C) would be required to determine their reactivity. Second, when multiple pairwise reactions take place within the temperature increments over which we sample ($\Delta T = 100$ °C), the algorithm cannot determine the precise reaction sequence between the lower and upper temperatures (*e.g.*, between 600 and 700 °C). In principle, the

second limitation can be overcome by using a smaller temperature interval; however, doing so would also require more experiments.

We have also clarified the meaning of the white squares in **Fig. 4**. These changes are provided below.

From the caption of Fig. 4: White squares (unshaded) represent pairs of phases whose reactivity was not learned by ARROWS³.

Comment 7:

In the YBCO case, why is the number of syntheses necessary to find all 10 good yield conditions used as the standard for comparison between BO, GA, and the current method? From the viewpoint of improving the synthesis process, it should be sufficient if any one of the ten can be found in the shortest possible time. Does the current method still show superiority when evaluated with such a metric?

Response:

We agree that, in practice, it is sufficient to identify just one optimal synthesis route for a given target. However, we tasked each approach with finding all ten optimal routes for YBCO to improve the statistics of the results and ensure that ARROWS³ performed well not only by chance. We would also like to note that identifying multiple synthesis routes can sometimes be beneficial for practical applications, *e.g.*, providing options from which the most practical and cost-effective route can be chosen.

When applied to find just one optimal synthesis route, ARROWS³ requires only ten experimental iterations. In contrast, the techniques based on BO and GAs require on average 16 and 17 iterations, respectively, to accomplish the same task. In addition to their reduced efficiency, BO and GAs each show a substantial degree of variability in the time required to identify an optimal synthesis route. To illustrate this, we have added a plot (**Supplementary Fig. 2a**, copied below) that shows the number of experiments required to identify just one optimal synthesis route for YBCO when implementing BO and GAs with different random starting seeds. For comparison, we also show the results from two deterministic algorithms (D-optimal design and ARROWS³) in **Supplementary Fig. 2a**. Further details on the implementation of D-optimal design (which is newly added to the manuscript) are given in our response to Comment 8.

The large spread in the number of iterations required to identify one optimal route persists even when identifying all ten optimal routes, as shown by **Supplementary Fig. 2b** (copied below). Moreover, the performance of ARROWS³ is substantially better than all other algorithms when used to identify all ten optimal routes as it has been given more time to learn which reactions are most effective to produce YBCO. In contrast, its performance is only slightly better than the other algorithms when used to identify just one optimal route.

Supplementary Figure 2: Distributions showing the number of experimental iterations required to identify (a) at least one optimal synthesis route for YBCO, or (b) all ten optimal synthesis routes for YBCO. Results are categorized by the optimization algorithm used to identify these routes. In each violin plot, the embedded box extends from lower to upper quartiles of the distribution. Black dots are used to denote

the mean. Because BO and GAs are stochastic, the number of iterations required by each varies substantially depending on the random starting seed. In contrast, D-optimal design and ARROWS³ are both deterministic.

In addition to these newly created supplementary figures, the task of identifying just one optimal synthesis route for YBCO is now discussed in the main text. Please find the relevant text below.

Results, YBCO, Page 9: While in practice it would be sufficient to identify just one optimal synthesis procedure for a given target, tasking the algorithm with identifying *all* optimal procedures for YBCO allows us to showcase its ability to learn over many experimental iterations (**Supplementary Fig. 2**). It also reduces the likelihood that ARROWS³ discovers an optimal synthesis route by chance, thereby increasing our confidence in the performance of the algorithm.

Comment 8:

As the authors mention, the way information is fed to the predictive model in BO or GA seems unfair compared to the current method, and I did not feel the comparison itself was meaningful. Rather, I believe it is necessary to compare with experimental design methods such as the D-optimal design.

Response:

We thank the reviewer for this interesting suggestion. An implementation of D-optimal design has now been tested and its results are included in the manuscript. A revised version of **Fig. 3a** is copied on the next page (below). The details of this approach are included in **Supplementary Note 1**.

D-optimal design appears to perform quite well when used to propose few experiments, outperforming all other algorithms tested here. We suspect its effectiveness can be attributed to its ability to select diverse experimental parameters based on different (untested) precursors. However, the optima discovered by D-optimal design quickly levels off when it is used to propose many experiments. It underperforms ARROWS³ after 40 experiments, at which point our algorithm can glean information from the previous experimental outcomes and suggest improved precursor combinations.

We also note that the number of experiments to propose using methods such as D-optimal design is difficult to choose *a priori* – the number of samples that must be tested to identify one (or all ten) optimal synthesis routes is not known beforehand.

Fig. 3: Optimization results from the experimental YBCO synthesis dataset. **(a)** Number of optimal synthesis routes identified as a function of the experimental iterations required by ARROWS³, Bayesian Optimization (BO), a Genetic Algorithm (GA), and D-Optimal design (D-Opt). The dashed line represents the total number of optimal synthesis routes in the dataset.

To discuss D-optimal design, substantial revisions have been made throughout the manuscript. Please find those changes copied below. We have also added a new folder (named “Black-Box”) to our github repository containing all the code used to evaluate Bayesian optimization, genetic algorithms, and D-optimal design.

Results, YBCO, Page 9: As a baseline with which to compare the performance of ARROWS³ on the YBCO dataset, we applied D-optimal design with progressively larger sets of proposed experiments. This approach aims to select the experiments whose outcomes will be maximally informative⁴⁶ to a model that maps the input variables (precursors and temperature) onto the output (YBCO yield). Here we assume a linear relationship between the two (**Supplementary Note 1**).

Results, YBCO, Page 10: ARROWS³ successfully identified all 10 optimal routes from 87 experiments, which account for just 46% of the entire design space (spanning 188 experiments). D-optimal design, on the other hand, required 165 experiments to accomplish the same task. Though, it is worth noting that D-optimal design was quick to identify three optimal synthesis routes in the first 12 experiments.

ARROWS³, although slower to identify optimal routes in the early stages of optimization, eventually surpassed D-optimal design once it gathered sufficient information regarding the reactivity of various phases in the Y-Ba-Cu-O chemical space.

Discussion, Page 22: Efficient data collection in vast experimental domains is a longstanding challenge. Traditional approaches based on design of experiments^{52,53}, including the D-optimal design algorithm tested here, can aid in the selection of experiments that are most informative to model a quantity such as target yield. However, these methods can fall short when dealing with a particularly large search space or when the quantity of interest is sparsely valued. Both challenges exist in solid-state synthesis, where many precursor combinations are often available for a given target, most of which fail to produce that target in any measurable amount. This warrants the use of active learning algorithms that can efficiently navigate the search space by adapting from failed experiments. Here we evaluated the performance of two such methods, Bayesian optimization and genetic algorithms, when applied to optimize the synthesis of YBCO. While each is known to perform well on continuous variables such as time or temperature^{54,55}, our tests show that they fail on the discrete task of precursor selection.

Supplementary Note 1

D-optimal design was used as an initial benchmark with which to compare the performance of ARROWS³ on the YBCO synthesis dataset. This approach is commonly used in the Design of Experiments (DoE), and it is designed specifically to select the combination of experimental parameters that maximize the determinant of the information matrix. For a more detailed explanation of optimal design and the information matrix, we refer the reader to previous work¹. Here we perform D-optimal design under the assumption that the yield of our target phase (YBCO) is linearly related to the selection of precursors (P_i) and synthesis temperature (T) through some coefficients (c_i) that can be learned:

$$Yield = \sum_{i=0}^N c_i P_i + c_{N+1} T$$

Where P_i is represented using a one-hot encoding as outlined in **Supplementary Note 2** and N is equal to the number of available precursors (*e.g.*, 11 precursors for YBCO). All temperatures are normalized such that values between 600 and 900 °C are mapped onto values between 0 and 1. After building the information matrix for this model, the parameters that maximized its determinant were identified by using the CVXPY and CVXOPT packages within Python. The number of experiments proposed by this approach were progressively increased from one set of parameters up to 188 sets of parameters (*i.e.*, all

experiments available in the YBCO space). The number of optimal synthesis routes (yielding pure YBCO) contained within each batch of proposed experiments was identified and used to plot the gray curve shown in **Fig. 3** of the main text.

Comment 9:

In the case of NTMO, there is a description that extending the calcination time improves the yield because $\Delta G'$ is small. Is this decision made automatically by the current method? If not, the “Autonomous decision making” in the paper title feels exaggerated.

Response:

The decision to increase the calcination time was not made automatically by the algorithm in its current implementation. We would like to clarify that the main application of ARROWS³ is to optimize the selection of precursors for targeted synthesis. It is less well-suited to handle temperature and hold times, which themselves are numerical and therefore could be readily optimized using more standard algorithms such as Bayesian optimization.

Nevertheless, the reviewer raises an interesting point and we do believe that the current algorithm could be extended to make decisions regarding temperature and hold time. For example, in cases where a moderate target yield (~50%) is obtained from a set of precursors with a short hold time, ARROWS³ may check whether the remaining reactions needed to improve target's yield are thermodynamically favorable. If this is the case, then a longer hold time may be used to enable such reactions to take place.

The possibility of these additions to ARROWS³ are now discussed in the manuscript (see below).

Discussion, Page 24: Beyond the selection of optimal precursors, synthesis planning often requires the heating profile to be carefully designed. Previous work has addressed this challenge by using standard optimization techniques⁵⁵, which perform well as the heating profile can be described in terms of continuous variables (*e.g.*, temperature and time). However, our findings show that a more physics-informed approach may also be viable. For the synthesis of each target material studied in this work, ARROWS³ used a short hold time (4 h) to identify the precursors and temperature that give maximal target yield. If necessary, manual decisions were made to increase the hold time if 1) the target yield was lower than desired, and 2) the leftover reactions needed to grow the target were thermodynamically

favorable. In doing so, > 90% yield was obtained for all three target materials we considered. Moreover, it was shown that such high yield was possible only for the precursor sets optimized by ARROWS³ at short hold time, thereby demonstrating that long hold times need not be used when testing various precursors. Decisions regarding when to extend the hold time after identifying an optimal set of precursors could later be incorporated into ARROWS³, enabling further progress toward complete autonomy in solid-state synthesis.

Comment 10:

The chemical reaction formulas annotated on the XRD profile in Figure 1b need an explanation as to whether they indicate a combination of precursors or phases identified from XRD.

Response:

The chemical formulae presented in **Fig. 1b** correspond to phases identified from XRD performed on samples heated at the specified temperatures. This is now clarified in the figure, as well as its caption.

From the caption of Fig. 1: (b) Experiments are performed at iteratively higher temperatures to identify reaction intermediates. **The chemical formulae listed in this panel represent phases identified from XRD measurements at each temperature.**

Comment 11:

The temperature is inconsistent between the caption of Figure 5a and the main text.

Response:

This error has now been corrected such that the caption mentions the correct temperature (400 °C).

From the caption of Fig. 5: The top panel shows the weight fraction of NTMO obtained from each precursor set when **tested at 400 °C**. The bottom panel displays the weight fraction of a competing phase, Na₂Mo₂O₇, **obtained at 300 °C**.

Reviewer #3

General Assessment:

The focus of this manuscript is the development of an autonomous decision-making approach for inorganic solid-state syntheses. The use of high temperature solid state syntheses has, of course, results in a host of technologically important materials. While the mechanical steps of this synthetic approach appear simple, a great deal of complexity lies under the surface. The authors nicely focus on an often unobserved, but critically important component of these reactions, the formation of intermediate phase that either enable or preclude the formation of the desired product.

Response:

We thank the reviewer for their feedback. Each of their comments are addressed below.

Comment 1:

A central question that remains in the mind of this reviewer is the construction of the initial reaction parameters. These include, temperature ranges, hold times, the presence or absence of regrinding steps, and reaction conditions (oxidizing, reducing, inert atmospheres).

- The authors note that hold times are specified for each study. The chosen times are rather short (4 or 8 h) with respect to more tradition high temperature solid state approaches. How were these times chosen? The choices of reaction time between regrinding steps more often conform to experimenter schedules than the reactions themselves (there is nothing magical about 24 h), and shorter times are desired of course. Having said that, there is some concern that short times without repeated regrinding steps selects for kinetic products in reactions generally governed by thermodynamics.
- The chemistry of many solid-state reactions can and does vary as a function of temperature. By this I mean that the phase diagram being explored does not remain unchanged as the temperature increases or decreases. For example, the authors correctly note that the traditional synthesis temperature for YBCO lies above the temperature window explored in this work. The identification of BaCO₃ as a problematic reactant at lower temperatures does not map onto its demonstrated utility at 950 C.

Response:

Regarding the first comment from the reviewer, we chose to use short hold times specifically to make the problems more difficult, *i.e.*, to provide more challenging test cases on which to validate our algorithm. For example, YBCO is generally thought to be thermodynamically stable at high temperature, and therefore any combinations of precursors would likely give high yield if held long enough at high temperature and subjected to intermittent regrinding. However, our work aims to show that ARROWS³ can identify *fast* reaction pathways enabled by large ΔG , which can be achieved through optimal precursor selection. We believe this to be of interest for commercial applications in the large-scale synthesis of materials, where shorter hold times can assist in conserving energy and costs.

Furthermore, we believe that experiments with short hold times can act as surrogates to identify precursor combinations that are most effective, even when subjected to improved grinding longer hold times. Indeed, this was demonstrated for two of the targets considered in our work. An optimal precursor set (Na₂O, MoO₃, and TeO₂) for NTMO was identified using experiments with a hold time of only 4 h. When this same set was heated for a longer time of 8 h, the target yield increased from 62% to 94%. In contrast, when a non-optimal set was tested using a longer hold time of 8 h, it did not produce any detectable amount of NTMO.

We have now clarified these points in the manuscript. The possibility of extending automated decision making to modify the heating profile is also discussed. The relevant text is provided below.

Results, YBCO, Page 8: Such a short hold time was used specifically to make the optimization task more challenging, as longer heating durations with intermittent regrinding are typically required to form highly pure YBCO samples⁴².

Discussion, Page 24: Beyond the selection of optimal precursors, synthesis planning often requires the heating profile to be carefully designed. Previous work has addressed this challenge by using standard optimization techniques⁵⁵, which perform well as the heating profile can be described in terms of continuous variables (*e.g.*, temperature and time). However, our findings show that a more physics-informed approach may also be viable. For the synthesis of each target material studied in this work, ARROWS³ used a short hold time (4 h) to identify the precursors and temperature that give maximal target yield. If necessary, manual decisions were made to increase the hold time if 1) the target yield was

lower than desired, and 2) the leftover reactions needed to grow the target were thermodynamically favorable. In doing so, > 90% yield was obtained for all three target materials we considered. Moreover, it was shown that such high yield was possible only for the precursor sets optimized by ARROWS³ at short hold time, thereby demonstrating that long hold times need not be used when testing various precursors. Decisions regarding when to extend the hold time after identifying an optimal set of precursors could later be incorporated into ARROWS³, enabling further progress toward complete autonomy in solid-state synthesis.

Regarding the second comment from the reviewer, we agree that temperature can have a significant impact on synthesis outcomes. Generally, prior knowledge should be employed in designing the search space over which ARROWS³ performs its optimization (please see our response to Comment 4 from Reviewer #2). In the case of YBCO, we chose to keep the synthesis temperatures low specifically to make the problem more challenging for ARROWS³, as well as to avoid melting, which we found made sample extraction more difficult.

The selection of temperature bounds is now discussed in more detail in the **Methods** (see below).

Methods, Formulation of the search space, Page 26: Any prior knowledge regarding the chemical system should be used when designing the search space. For example, the lower temperature bound (T_{\min}) may be chosen to exceed the known decomposition temperatures of all carbonates and hydroxides being considered as precursors. Similarly, the upper temperature bound (T_{\max}) may be chosen below the melting points of the precursors if the user wishes to retain a product consisting of solid powder. With respect to precursor selection, it may often be beneficial to exclude compounds that are known to be inert in the temperature range being considered; however, this can also be learned by ARROWS³ through experimentation (see **YBCO** in the main text).

Comment 2:

A second central theme is a quest to obtain routes for phase pure samples of known or unknown materials as quickly as possible. The infrastructure balance has shifted in this proposed synthetic methodology from traditional approaches. Traditionally one would run many reactions in parallel in a single furnace, as composition can vary between samples, but temperature cannot. Conducting a range of reagent combinations at different temperatures either requires more furnaces or more sequential studies. The

authors should comment on this balance.

Response:

Please refer to our response to Comment 1 from Reviewer #2. We have now performed additional tests that illustrate the role of parallel experiments in optimization campaigns guided by ARROWS³.

Comment 3:

The possibility of transfer learning in this approach is interesting and enticing. The authors note the possibility of predicting the formation of other compounds within the Y-Ba-Cu-O system (or more accurately – compound that could be made from the reagents used in the YBCO study). Does transfer learning extend between systems? If, for example, one was to explore BSCCO, would the YBCO work transfer?

Response:

The pairwise reactions learned by ARROWS³ are transferrable so long as there exists some overlap between the new and previously tested chemistries. In other words, some elements (and their precursors) must be shared between the two target materials being considered.

The degree to which transfer learning will be successful is defined by the *amount* of overlap between the two chemical spaces, *i.e.*, how many elements are shared between them. In the example provided by the reviewer (Bi-Sr-Ca-Cu-O), only one of the metals (Cu) is shared with chemical space tested in this work (Y-Ba-Cu-O). As such, only reactions involving Cu (*e.g.*, thermal decomposition of CuCO₃) will be transferred when performing optimization in the BSCCO space.

If, on the other hand, two or more elements are shared between chemical spaces, then the pairwise reactions involving those elements will be transferred.

We now clarify this point in the **Discussion**.

Discussion, Page 23: Predictions of this nature will in general become more abundant as the overlap between chemical spaces increases, specifically when considering target materials with two or more shared elements.

Comment 4:

The identification of pairwise interactions that result in either reactive or inert intermediates is both important and informative.

Response:

We thank the reviewer for their positive feedback.

Comment 5:

The authors should be cautious in the determination of weight percent using diffraction data. Many parameters affect peak intensity (or area) that are distinct from weight percent. These include grain size, crystallinity, average atomic scattering factor to name three.

Response:

We agree with the reviewer that obtaining precise weight fractions is a complicated task that requires careful considerations of many factors. In our tests, we only estimated the weight fractions such that they may be compared in a relative fashion, *i.e.*, a reaction with high target yield can be distinguished from one with low target yield.

We have now clarified these points in the manuscript (see below).

Methods, YBCO synthesis, Page 30: Weight fractions were approximated by assessing the relative peak intensities of the constituent phases in each mixture. A more careful approach based on Rietveld refinement, which accounts for properties such as grain size and texture, would be required to obtain precise weight fractions. However, this work only requires that we compare relative weight fractions between different experiments, enabling ARROWS³ to identify the most effective synthesis route for a given target.

REVIEWER COMMENTS

Reviewer #2 (Remarks to the Author):

I have reviewed the thoughtful response from the Authors. I believe the quality of the manuscript, which was already high, has been further enhanced. However, my comment that this paper lacks impact for synthesis researchers remains unchanged after reading the revised manuscript. This is because synthesis researchers believe that it would be faster and more reliable to conduct a few hundred synthesis experiments rather than using this method to sequentially carry out experiments when the target compound is determined and the corresponding starting materials are listed. I think this is due to a history where the thermodynamic stability calculated by first-principles calculations has not necessarily led to synthesizability. Also, the lack of surprises in the chemical trends obtained by this method, such as issues with BaCO₃ or MoO₂, may be a contributing factor. I thought that in the future, problems such as kinetics that are not provided by first-principles calculation results might be further improved by devising ways to feed back experimental results. Furthermore, I think that the more complex the system, the greater the benefits of this method.

As I wrote in my previous comments, this paper concisely summarizes the Authors' ideas and the actual process that underlies this method. Apart from the point that there is no impact on synthesis researchers, this revision has further improved, and I think it has become a good manuscript. Also, I have one question regarding the newly added part.

I have a question about the exploration with different batch sizes shown in Supplementary Figure 4a. In the case of a batch size of 16, the increment in the No. of Samples Tested should be 16, but looking at the graph, it seems to be increasing by a smaller number each time. Also, why is there no difference due to batch size in the initial rise?

Reviewer #3 (Remarks to the Author):

The authors have increased the quality and clarity of this manuscript through the revisions they describe. This manuscript is quite excellent and I believe that the science it contains represents a step forward. The strategy to bring data science and ML techniques to bear upon the selection of reactants (based upon both desirable and undesirable combinations) is important. The role of parallelization is important in such experimental efforts, as the experimentalist are focused on minimizing the number of reactions (reagent cost, experimenter effort) and wall time. These two factors are often at odds with one another, and the ability to specify a batch size is a welcome addition. The data presented in Supplementary Figure 4a are interesting. The presentation of these data highlights the value in more batches (for a given total number of reactions, the number of optima discovered is highest for smaller batch sizes (more opportunities to learn between batches)). However, given fixed hold times for each reaction, the authors should also plot the number of optima discovered vs wall time. On page 13, the authors describe 23 unshaded (white) squares in Figure 3. These squares are present in Figure 4, not Figure 3.

Reviewer #2

General Assessment:

I have reviewed the thoughtful response from the Authors. I believe the quality of the manuscript, which was already high, has been further enhanced. However, my comment that this paper lacks impact for synthesis researchers remains unchanged after reading the revised manuscript. This is because synthesis researchers believe that it would be faster and more reliable to conduct a few hundred synthesis experiments rather than using this method to sequentially carry out experiments when the target compound is determined, and the corresponding starting materials are listed. I think this is due to a history where the thermodynamic stability calculated by first-principles calculations has not necessarily led to synthesizability. Also, the lack of surprises in the chemical trends obtained by this method, such as issues with BaCO₃ or MoO₂, may be a contributing factor. I thought that in the future, problems such as kinetics that are not provided by first-principles calculation results might be further improved by devising ways to feedback experimental results. Furthermore, I think that the more complex the system, the greater the benefits of this method.

As I wrote in my previous comments, this paper concisely summarizes the Authors' ideas and the actual process that underlies this method. Apart from the point that there is no impact on synthesis researchers, this revision has further improved, and I think it has become a good manuscript.

Comment 1:

I have a question about the exploration with different batch sizes shown in Supplementary Figure 4a. In the case of a batch size of 16, the increment in the No. of Samples Tested should be 16, but looking at the graph, it seems to be increasing by a smaller number each time.

Response:

In the original version of Supplementary Fig. 4a, we simulated experimental batching by restricting our decision-making algorithm from updating the order in which samples were tested until after every N experiments, with N being the batch size. However, when tracking the number of optimal synthesis routes discovered, we still examined each sample individually based on their order within each batch. As such, the curves plotted in Supplementary Fig. 4a could rise more frequently than the designated batch size.

We agree with the reviewer that our original approach to the simulation of experimental batching may be difficult to follow, and is not necessarily representative of actual experiments, where new optimal synthesis routes can only be discovered after a full batch of samples have been tested. Accordingly, we have now updated the Supplementary Fig. 4a such that the number of optima can increase only when the sample number is a multiple of the designated batch size. The update figure is copied below, in addition to the new panels (b and d) that were created in response to Reviewer 3's comments.

Supplementary Figure 4: Effects of batch size on the optimization of YBCO synthesis. (a) Number of optimal synthesis routes identified with respect to the number of samples queried by ARROWS³. Each curve represents an optimization campaign performed with a distinct batch size. (b) Number of optimal synthesis routes discovered versus the furnace hold for evaluating the required number of batches. (c) Number of batches and samples required to identify all ten optimal synthesis routes, with each dot symbolizing an optimization campaign for a specific batch size. (d) Total furnace hold time required to identify all optimal synthesis routes for each batch size.

Comment 2:

In Supplementary Figure 4a, why is there no difference due to batch size in the initial rise?

Response:

The initial experiments show little change because our decision-making algorithm requires some time to gather enough data before adjusting its priority of experiments. When few pairwise reactions have been observed, the algorithm does not have enough information to determine which precursor sets will produce favorable intermediates that retain a large driving force to form the desired target. Without the ability to predict the outcomes of new (untested) precursor sets, no changes are made to the order in which experiments are performed. This means that changing the batch size will have little effect on the experimental sequence early on, and therefore the curves are unaffected at low sample number. In contrast, many updates are made at later stages. Because increasing the batch size reduces the opportunity window to make these updates, it has a large effect on the resulting curves in Supplementary Fig. 4a. This point is now clarified in the main text.

From Page 15: The efficiency with which samples are queried becomes particularly affected at later stages in the experiments, where the algorithm has sufficient knowledge of the chemical space to make frequent updates to its ranking of different precursor sets.

Reviewer #3

General Assessment:

The authors have increased the quality and clarity of this manuscript through the revisions they describe. This manuscript is quite excellent, and I believe that the science it contains represents a step forward. The strategy to bring data science and ML techniques to bear upon the selection of reactants (based upon both desirable and undesirable combinations) is important. The role of parallelization is important in such experimental efforts, as the experimentalist are focused on minimizing the number of reactions (reagent cost, experimenter effort) and wall time. These two factors are often at odds with one another, and the ability to specify a batch size is a welcome addition. The data presented in Supplementary Figure 4a are interesting. The presentation of these data highlights the value in more batches (for a given total number of reactions, the number of optima discovered is highest for smaller batch sizes (more opportunities to learn between batches)).

Response:

We thank the reviewer for their positive feedback.

Comment 1:

Given fixed hold times for each reaction, the authors should also plot the number of optima discovered versus wall time.

Response:

Two new plots analyzing the effect of batch size on the total furnace hold time required to identify optimal synthesis routes have now been added to Supplementary Fig. 4 (panels **b** and **d**). The revised figure is also copied below.

Supplementary Figure 4: Effects of batch size on the optimization of YBCO synthesis. **(a)** Number of optimal synthesis routes identified with respect to the number of samples queried by ARROWS³. Each curve represents an optimization campaign performed with a distinct batch size. **(b)** Number of optimal synthesis routes discovered versus the furnace hold for evaluating the required number of batches. **(c)** Number of batches and samples required to identify all ten optimal synthesis routes, with each dot symbolizing an optimization campaign for a specific batch size. **(d)** Total furnace hold time required to identify all optimal synthesis routes for each batch size.

Comment 2:

On page 13, the authors describe 23 unshaded (white) squares in Figure 3. These squares are present in Figure 4, not Figure 3.

Response:

The text has been corrected so that it now refers to Figure 4.